# Factors affecting the implementation of calcium supplementation strategies during pregnancy to prevent pre-eclampsia: a mixed-methods systematic review

Gabriela Cormick [1,2] Hellen Moraa,[3] Rana Islamiah Zahroh,[4] John Allotey,[5,6] Thaís Rocha,[5] Juan Pablo Peña-Rosas,[7] Zahida P Qureshi,[3] G Justus Hofmeyr,[8,9] Hema Mistry,[10] Luc Smits,[11] Joshua Peter Vogel,[12] Alfredo Palacios,[13,14] George N Gwako,[3] Edgardo Abalos,[15] Koiwah Koi Larbi,[16] Guillermo Carroli,[17] Richard Riley,[5] Kym IE Snell,[18] Anna Thorson,[19] Taryn Young,[20] Ana Pilar Betran,[21] Shakila Thangaratinam,[5,6,22] Meghan A Bohren [4]

For numbered affiliations see end of article.

**Correspondence to**
Dr Gabriela Cormick;
gabmick@yahoo.co.uk

## ABSTRACT

**Objectives** Daily calcium supplements are recommended for pregnant women from 20 weeks' gestation to prevent pre-eclampsia in populations with low dietary calcium intake. We aimed to improve understanding of barriers and facilitators for calcium supplement intake during pregnancy to prevent pre-eclampsia.

**Design** Mixed-method systematic review, with confidence assessed using the Grading of Recommendations, Assessment, Development and Evaluations-Confidence in the Evidence from Reviews of Qualitative research approach.

**Data sources** MEDLINE and EMBASE (via Ovid), CINAHL and Global Health (via EBSCO) and grey literature databases were searched up to 17 September 2022.

**Eligibility criteria** We included primary qualitative, quantitative and mixed-methods studies reporting implementation or use of calcium supplements during pregnancy, excluding calcium fortification and non-primary studies. No restrictions were imposed on settings, language or publication date.

**Data extraction and synthesis** Two independent reviewers extracted data and assessed risk of bias. We analysed the qualitative data using thematic synthesis, and quantitative findings were thematically mapped to qualitative findings. We then mapped the results to behavioural change frameworks to identify barriers and facilitators.

**Results** Eighteen reports from nine studies were included in this review. Women reported barriers to consuming calcium supplements included limited knowledge about calcium supplements and pre-eclampsia, fears and experiences of side effects, varying preferences for tablets, dosing, working schedules, being away from home and taking other supplements. Receiving information regarding pre-eclampsia and safety of calcium supplement use from reliable sources, alternative dosing options, supplement

## STRENGTHS AND LIMITATIONS OF THIS STUDY

⇒ We adopted a mixed-methods approach, which allowed for inclusion and consolidation of studies with a range of designs.
⇒ The strength of our review lies on behaviour change frameworks mapping, which improved our understanding of how barriers and facilitators influence calcium uptake and potential strategies to address them.
⇒ The transferability of our results may be limited, as studies were mostly from low-income and middle-income countries and almost half came from the same project in Kenya and Ethiopia.
⇒ Most included studies engaged women who attended antenatal care, therefore, might not be representative of those that do not reach the health system during pregnancy.

reminders, early antenatal care, free supplements and support from families and communities were reported as facilitators. Healthcare providers felt that consistent messaging about benefits and risks of calcium, training, and ensuring adequate staffing and calcium supply is available would be able to help them in promoting calcium.

**Conclusion** Relevant stakeholders should consider the identified barriers and facilitators when formulating interventions and policies on calcium supplement use. These review findings can inform implementation to ensure effective and equitable provision and scale-up of calcium interventions.

**PROSPERO registration number** CRD42021239143.

## INTRODUCTION

Hypertensive disorders of pregnancy are among the leading causes of maternal

and perinatal morbidity and mortality globally.[1] Pre-eclampsia is a hypertensive disorder of pregnancy characterised by hypertension developing after 20 weeks' gestation, combined with proteinuria or other new onset of maternal organ dysfunction, while eclampsia is a severe form of pre-eclampsia characterised by seizures.[2 3] Pre-eclampsia contributes to approximately 14% of the 300 000 maternal deaths worldwide annually.[4] Management of pre-eclampsia requires regular monitoring and evaluation of the woman and her baby to achieve an optimal timing of birth and prevent severe complications.[5] Preventive strategies are essential to reduce the burden of morbidity and mortality, especially in low-income and middle-income countries (LMICs) where most complications occur.

The WHO recommends 1.5–2 g a day from 20 weeks' gestation for women who are living in populations with low dietary calcium intake, especially those at high risk of developing pre-eclampsia.[6] This is aligned with findings from a systematic review that reported calcium supplements during pregnancy compared with placebo may reduce the risk of pre-eclampsia by 55% (13 trials, 15 730 participants; relative risk (RR) 0.45, 95% CI 0.31 to 0.65).[7] Moreover, maternal death or severe morbidity was reduced by 20% with calcium supplements (4 trials, 9732 participants; RR 0.80, 95% CI 0.66 to 0.98).[7] The evidence base has since been updated multiple times, with WHO recommendation consistently updated up to 2018.[8–10]

Despite the WHO recommendation, calcium supplement during pregnancy remains low in LMICs and rates of pre-eclampsia are not falling in regions where calcium supplementation is recommended.[10] Practical challenges to implementing WHO recommendations have been documented. For example, women need to take three-spaced tablets to achieve the requisite daily dose, and this needs to be separated from timing of intake of other supplements (such as iron) to optimise calcium absorption.[11] In addition, antenatal care services need consistent supplies of calcium tablets, which can be hindered by logistical issues in supplement distribution and storage.[7] We conducted a mixed-methods systematic review aiming to improve understanding of the barriers and facilitators of calcium supplement intake during pregnancy to prevent pre-eclampsia, from the perspectives of women, families, community members, healthcare providers and policy-makers.

## METHODS

This mixed-methods review is reported according to Preferred Reporting Items for Systematic Reviews and Meta-Analyses (online supplemental appendix 1),[12] Enhancing Transparency in Reporting the Synthesis of Qualitative Research statement (online supplemental appendix 2),[13] and based on guidance from Cochrane Effective Practice and Organisation of Care.[14] The protocol was registered on PROSPERO (CRD42021239143).

### Topic of interest and types of studies

We included studies that documented perspectives, perceptions and experiences of women who experienced or were at risk of pre-eclampsia and/or received calcium-containing supplements. We also included studies on the views of their partners or families, as well as studies on maternity healthcare providers (eg, midwives, nurses, doctors) and other relevant stakeholders (eg, facility managers, policy-makers) involved in decisions on calcium supplements in pregnancy. There were no limitations imposed on geographical location or type of health facility. The time frame for using calcium supplement is during pregnancy, independent of the gestational age.

We included primary qualitative, quantitative and mixed-methods studies reporting implementation or use of calcium supplements in any presentations including powder, granule, chewable tablet, capsule, liquid-filled capsule, tablet, suspension or powder for suspension. We did not include studies assessing the effects of calcium-fortified foods or beverages. We excluded case reports or case series, letters, editorials, commentaries, reviews, study protocols, posters and conference abstracts or other study sources that did not provide primary data.

### Search strategy

We searched MEDLINE and EMBASE via Ovid, CINAHL and Global Health via EBSCO to identify eligible studies from inception to 17 September 2022. A search strategy was developed and adapted for each database (online supplemental appendix 3), using different terms for calcium and pregnancy. No limitations on publication date or language were imposed. Grey literature searches were also conducted using OpenGrey and Google. We checked reference lists of included studies to identify any relevant records not retrieved in the database search.

### Study selection

After removing duplicates in EndNote, records were imported to Covidence for screening.[15] Two of the following authors (GCormick, GNG, AP, RIZ and HM) independently assessed eligibility of each record by comparing titles and abstracts against the eligibility criteria. Full texts of potentially eligible papers were retrieved and assessed, disagreements were resolved through discussion or consulting a third author. Papers emerging from the same study were collated and treated as one data source. Titles and abstracts of papers published in languages other than English, French and Spanish were translated through open-source software (Google Translate) to assess their eligibility. Had we identified any relevant titles or abstracts in languages other than English, French and Spanish, we would have sought formal translation of the full texts from a native speaker.

### Data extraction and assessment of methodological limitations

Using a predesigned form, two reviewers (GCormick and RIZ) independently extracted data from included studies on study characteristics (setting, sample size,

characteristics of participants and objectives), design (data collection and analysis methods), qualitative data (themes, findings and quotations) and quantitative data (data source, outcome measures, results and measures of compliance or uptake).

All included studies underwent quality appraisal by two authors (GCormick, HM and RIZ). As the review included quantitative, qualitative and mixed-methods studies, we used the Mixed Methods Appraisal Tool, which produces a single quality rating on the basis of: aims, methodology, design, recruitment, data collection, blinding, data analysis, selective reporting, reflexivity, ethical considerations, results, research contribution and other sources of bias.[16] Appraisal of study quality was used to inform data analysis, and not to exclude studies.

## Quality appraisal, analysis and assessing confidence

We conducted a preliminary qualitative synthesis using a thematic analysis approach.[17] Thematic analysis is a valuable method in analysing qualitative data to examine perspectives, preferences, experiences, acceptability, feasibility and other factors that can influence implementation.[17] The analysis begins with initial readings to build our familiarity with the data. Two reviewers (GCormick and HM) independently conducted line-by-line coding of findings of two qualitative studies.[18 19] From this, we developed the qualitative codebook, which was then used to code all other included studies. Next, we generated analytical themes and interpretations to explore relationships within and across studies. This was achieved by organising codes into a hierarchy and identifying barriers and facilitators between study characteristics and findings or exploring different findings across studies. Once qualitative themes were generated, a summary of qualitative findings was developed. Quantitative findings were then narratively mapped to qualitative themes to explore areas of convergence or divergence. ATLAS.ti was used to manage data analysis.

After the thematic analysis, we mapped the qualitative themes to the Theoretical Domains Framework (TDF) and Capability, Opportunity and Motivation of Behaviour (COM-B) models.[20 21] TDF and COM-B are inter-related behaviour change models which can guide implementation research and intervention design in understanding barriers and facilitators of intended behaviours. We used TDF and COM-B to explore barriers and facilitators of healthcare providers and women to use calcium supplements during pregnancy using evidence-based behaviours to identify potential behaviour change intervention strategies.

We used the GRADE-CERQual (Grading of Recommendations, Assessment, Development and Evaluations-Confidence in the Evidence from Reviews of Qualitative research) approach to assess confidence in each qualitative finding.[22] GRADE-CERQual assessed confidence based on four key components: methodological limitations,[23] coherence,[24] adequacy[25] and relevance.[26] After assessing each of the four components, we assessed the overall confidence[22] as high, moderate, low or very low.

## Patient and public involvement
None.

## RESULTS
We included 18 papers from 16 studies (figure 1). The included papers were published in English between 2014 and 2022. Out of 16 studies, 10 were quantitative and came from 11 papers,[27–37] 4 were qualitative[18 19 38 39] and 2 were mixed-methods and came from 3 papers.[40–42] Detailed study characteristics can be found in table 1.

Five studies aimed to evaluate the implementation of calcium supplements in pregnancy.[27 30 31 39 40] Seven studies evaluated the incorporation of calcium supplement recommendation to other recommended supplements taken during pregnancy, including aspirin,[18] and iron and folic acid supplements.[19 28 33 37 38 41 42] Four studies focused on general nutritional practices during pregnancy[29 32 34 35] and one study on all types of micronutrient supplements used before and during pregnancy.[36] Five studies came from one project, Micronutrient Initiative-Cornell University Calcium projects[19 28 33 38–41] and four studies came from Alive & Thrive project.[32 34 35 42]

The studies were conducted in seven different countries across three regions. In sub-Saharan Africa, three studies were conducted in Kenya,[28 33 38 39] one study in Ethiopia[19] and one study in Kenya in Ethiopia.[40 41] In Asia, four studies were conducted in India,[27 29 32 42] three studies in Bangladesh,[30 34 35] two studies in Nepal[31 37] and one study in China.[36] There is one study conducted in Europe, specifically the Netherlands.[18]

All four qualitative studies involved pregnant women and healthcare providers,[18 19 38] while one study also included adherence partners to support and remind women to take medication.[39] One mixed-method study included pregnant women[40 41] and one included healthcare providers and facility staff.[42] Among the 10 quantitative studies, 8 included pregnant women or women who had recently given birth,[27–30 32–34 36 37] 1 included healthcare providers[31] and 1 included both women and their husbands.[35]

Results of the critical appraisal of the included studies are available in online supplemental appendix 4. The main areas of concern for qualitative studies were an unclear or partial description of reflexivity and limited information regarding ethical considerations. For the quantitative studies, main concerns were regarding the appropriateness of measurement tools, lack of detail regarding non-response bias and insufficient information regarding statistical analysis.

## Qualitative and quantitative synthesis
We identified five themes related to factors affecting calcium supplement intake during pregnancy: (1) women's existing knowledge and learning; (2) women's

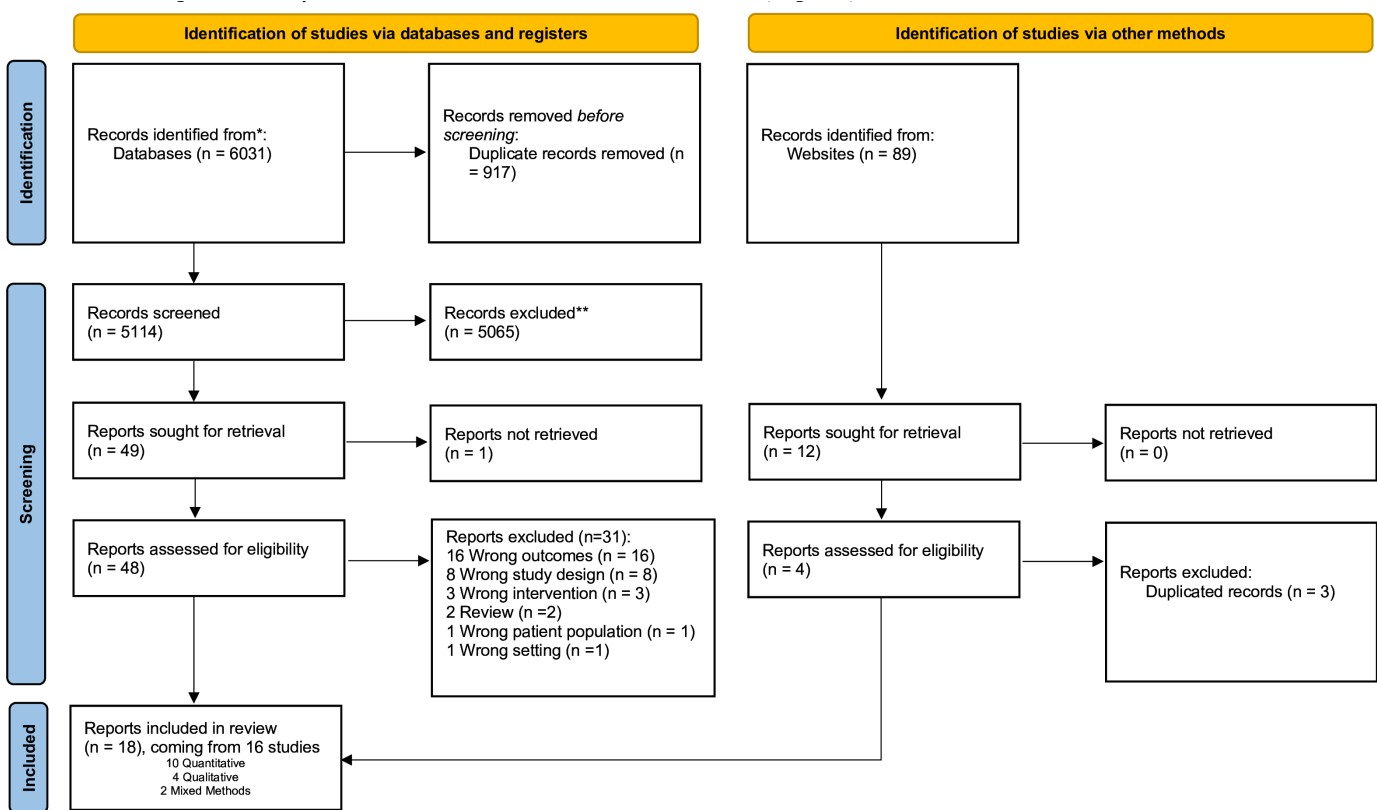

**Figure 1** PRISMA flow diagram. PRISMA flow chart illustrating the number of records included and excluded at various screening and reviewing steps, leading to final list of records for data extraction and meta-analysis.*Consider, if feasible to do so, reporting the number of records identified from each database or register searched (rather than the total number across all databases/registers). **If automation tools were used, indicate how many records were excluded by a human and how many were excluded by automation tools. PRISMA, Preferred Reporting Items for Systematic Reviews and Meta-Analyses.

beliefs about calcium supplements; (3) calcium supplement characteristics and dose regimens; (4) challenges due to daily routines and food insecurity and (5) strategies to improve the use of calcium. We also identified three themes related to factors affecting healthcare providers' prescription of calcium supplements: (1) health provider knowledge and training; (2) their beliefs about calcium supplements and (3) structural factors on site. Across all themes, there were 19 qualitative findings (table 2): 11 findings were high confidence, 6 were moderate confidence and 2 were low confidence (online supplemental appendix 5: evidence profile). The quantitative findings which were mapped to qualitative themes can be found in online supplemental appendix 6.

### Women's knowledge and learning
#### Women's knowledge about pre-eclampsia
Most women had limited knowledge about pre-eclampsia, and these conditions were not typically considered by women to be a serious problem. Symptoms of pre-eclampsia such as swollen feet, severe headache, blurred vision and vomiting were considered normal signs and symptoms in pregnancy, while seizures were associated with evil attacks or nutritional deficiencies (Finding 1.1—High confidence).[18 19 38] Women and healthcare providers from Ethiopia and Kenya stated that there was no local language for pre-eclampsia, which made it difficult for healthcare providers to explain the condition to women and served as a barrier in providing adequate knowledge to women and encouraging them to use calcium supplements.[19 38]

#### Information provision to women
Women felt they did not receive adequate information during pregnancy from healthcare providers about pre-eclampsia and calcium supplements. They described wanting to be given more information, regardless of their pre-eclampsia risk status. Women believed that having this essential information could help them to make informed choices and actively participate in their care. There were, however, mixed opinions from healthcare providers—some feared that more information could generate anxiety for women, while others were more supportive of providing information (Finding 1.2—Low confidence).[18 39] Healthcare providers were worried that sharing information about pre-eclampsia, especially with low-risk pregnant women, could lead to anxiety and make an 'uncomplicated pregnancy more stressful'.[18] Healthcare providers viewed their role as informants, but not as decision-makers for women. They believed that it should be a woman's choice to decide whether to consume calcium supplements or not.[18] Women and

**Table 1** Characteristics of included studies

| Study | Country | Population | No of participants | Methods | Ref |
|---|---|---|---|---|---|
| Qualitative | | | | | |
| Birhanu 2018 | Ethiopia | Pregnant women and healthcare providers | 20 women, 22 healthcare providers | Phenomology | [19] |
| Martin 2017a | Kenya | Pregnant women and healthcare providers | 22 women, 20 healthcare providers | Phenomology | [38] |
| Martin 2018 | Kenya | Healthcare providers, pregnant women, adherence partners | 7 healthcare providers, 32 pregnant women, 20 adherence partners | Phenomology | [39] |
| Vestering 2019 | Netherlands | Healthcare providers and pregnant women | 8 healthcare providers, 25 women | Phenomology | [18] |
| Mixed methods | | | | | |
| Kachwaha 2022 | India | Healthcare providers, supervisors and facility staffs | ~500 healthcare providers and supervisors, and 20 block level staffs | Mixed methods | [42] |
| Martin 2017b and Omatoyo 2018a | Kenya and Ethiopia | Pregnant women | 85 pregnant women, 50 in Ethiopia, 38 in Kenya | Trials of improved practices | [40 41] |
| Quantitative | | | | | |
| Baxter 2014 | Bangladesh | Pregnant women | 132 | Modified discrete-choice trial | [30] |
| Bora 2022 | India | Pregnant women | 3 097 274 | Cross-sectional study | [27] |
| Ghosh-Jareth 2015 | India | Pregnant women and recently delivered women | 184 pregnant women and 160 recently delivered women | Cross-sectional study | [29] |
| Liu 2019 | China | Women aged from 16 to 49 years who were pregnant between 2010 and 2013 and had specific pregnancy outcomes before the survey | 30 027 | Cross-sectional study | [36] |
| Martin 2017c and Omatoyo 2018b | Kenya | Pregnant women | 990 pregnant women and 16 facilities | Process evaluation study adopting programme impact pathway | [28 33] |
| Nguyen 2017 | Bangladesh | Pregnant women and recently delivered women | 600 pregnant women and 2000 recently delivered women | Cross-sectional household surveys | [34] |
| Nguyen 2018 | Bangladesh | Women and husbands | 1000 women and 70% of their husbands | Cluster randomised control trial with cross-sectional household surveys | [35] |
| Nguyen 2019 | India | Pregnant women and recently delivered women | 667 pregnant women and 1835 recently delivered women | Cross-sectional household surveys | [32] |
| Shakya Shrestha 2020 | Nepal | Pregnant women | 191 | Cross-sectional study | [37] |
| Thapa 2016 | Nepal | Pregnant and postpartum women, health facilities, female community health volunteers | 1240 | Prospective collection and secondary analysis of monitoring data captured by the MOHP | [31] |

healthcare providers mentioned that the scope of information provided to women should include symptoms of pre-eclampsia as well as effectiveness, benefits and safety of calcium-containing supplements.[18]

### Learning about calcium supplements

Women typically learnt about dietary calcium, including pre-eclampsia symptoms, from healthcare providers. They considered healthcare providers to be the most trusted and reliable source of information and reported feeling confident about taking calcium-containing supplements after receiving adequate information from them. Women also appreciated receiving information on calcium supplements and pre-eclampsia via information, education and communication (IEC) (materials such as) videos, media campaigns and trusted websites (Finding 1.3—High confidence).[18 19 32 34 39] Quantitative evidence supported qualitative findings regarding women's learning on calcium supplements. In the context of an intervention implementing training to healthcare providers to reinforce calcium-related messages to women, women reported that they would take calcium in a future pregnancy and that they would recommend calcium to other pregnant women.[31]

Women were more likely to consume calcium supplements if they had higher knowledge of calcium benefits (OR 11.7, 95% CI 5.97 to 22.86) and higher general education (OR 2.59, 95% CI 2.21 to 3.05).[32] Higher adherence was also reported in those with higher education (OR 1.45, 95% CI 1.31 to 1.60).[36] Higher nutrition knowledge was also associated with taking 6 times more calcium tablets.[32]

**Table 2** Summary of qualitative findings

| No | Summary of qualitative review findings | Contributing qualitative studies | Overall CERQual assessment | Explanation of overall assessment |
|---|---|---|---|---|
| Knowledge and learning | | | | |
| **1.1** | Women's knowledge about pre-eclampsia/eclampsia: Most women had limited knowledge about pre-eclampsia/eclampsia, and pre-eclampsia/eclampsia was not typically considered as a serious problem by women. General symptoms of pre-eclampsia/eclampsia such as swollen feet, severe headache, blurred vision and vomiting were considered normal in pregnancy, while seizures were linked to evil attacks or nutritional deficiencies | 26 27 34 | High confidence | Due to no or very minor concerns on coherence, minor concerns on methodological limitations (reflexivity and ethics), moderate concerns on relevance (1 out of 3 studies are indirectly relevant to our review aim and small number of countries), and minor concerns of adequacy (3 out of 6 contributed with 2 thick and 1 thin data) |
| 1.2 | Information provision to women: Women felt they did not receive adequate information during pregnancy from health workers about pre-eclampsia/eclampsia and calcium supplementation and would like to be given more information regardless of their risk status. Women believed that having this essential information could help them to make informed choices and to actively participate in their care. There were, however, mixed opinions from health workers, where some feared that more information could generate anxiety for women, while others were supportive of providing information to women | 13 34 | Low confidence | Minor concerns on methodological limitations (ethics and reflexivity), moderate concerns on coherence (no clear understanding on why some health workers worry in generating anxiety to women while the others are not), moderate concerns on relevance (small number of countries) and serious concerns on adequacy (2 out of 6 contributed with 1 thick and 1 thin data) |
| **1.3** | Learning about calcium supplementation: Women typically learnt about calcium, including pre-eclampsia and eclampsia symptoms, from health workers, and considered health workers as the most trusted and reliable source of information. They reported feeling confident about taking calcium after receiving adequate information from their health workers. Women also appreciated receiving information on calcium supplementation and pre-eclampsia/eclampsia from information, education and communication (IEC) (materials like) videos, media and trusted websites | 13 27 34 | High confidence | Due to no or very minor concerns on coherence, minor concerns on relevance (small number of countries), minor concerns on methodological limitations (reflexivity and ethics), and minor concerns on adequacy (3 out of 6 contributed, 1 moderate thick and 2 thin data) |
| Believe about the intervention | | | | |
| 2.1 | Fears about side effects as barriers to calcium uptake among women: Fears about side effects impact adherence to calcium consumption by women. Women highlighted that assurance of safe use of calcium is a key facilitator to consistent use. However, some women felt safety was not assured by health workers and were told by their families or communities that any pills consumed during pregnancy could be harmful, especially when the intervention was perceived as 'experimental' | 11 13 27 34 | High confidence | Due to no or very minor concerns on coherence, minor concerns on relevance (small number of countries), minor concerns on adequacy (small number of studies contributing to the qualitative evidence synthesis) and minor concerns on methodological limitations (reflexivity and ethics) |
| 2.2 | Experiences of side effects: Some women reported experiencing side effects after taking calcium and iron folic acid, which include dizziness, vomiting, nausea, stomach-ache, loss appetite, tiredness, diarrhoea, bloating and burping, yet noted that the side effects subsided with time. Women also reported to keep consuming the medication despite experiencing the side effects | 11 13 27 | High confidence | Due to no or very minor concerns on coherence, minor concerns on methodological limitations (ethics and reflexivity), moderate concerns on relevance (only low-income and lower-middle-income country and small number of countries) and minor concerns on adequacy (3 out of 6 studies contributed with 2 thick and 1 with thin data) |

Continued

**Table 2** Continued

| No | Summary of qualitative review findings | Contributing qualitative studies | Overall CERQual assessment | Explanation of overall assessment |
|---|---|---|---|---|
| 2.3 | Concerns of being stigmatised as HIV patient: Women expressed concerns about being stigmatised as HIV patients if they ingested calcium, which was a reported barrier to use. Some women were afraid of being stigmatised as their community often associated supplement consumption) and accompanying reminder posters with HIV | 13 34 | Moderate confidence | Due to no or very minor concerns on methodological limitations, no or very minor concerns on coherence, minor concerns on relevance (small number of countries) and serious concerns on adequacy (2 out of 6 study contributed) |
| 2.4 | Positive perceptions about calcium: Women reported that both their perceptions about expected benefits and previous experiences of taking calcium and iron-folic acid were facilitators of use. Women believed that consuming pills could compensate for suboptimal nutrition during pregnancy and that consuming nutrient supplementation during pregnancy would help keep the baby safe | 11 13 27 28 34 | High confidence | Due to no or very minor concerns on coherence, minor concerns on methodological limitations (ethics and reflexivity), minor concerns on relevance (1 study indirectly relevant to review aim and small number of countries) and adequacy (small number of studies contributing to the qualitative evidence synthesis) |
| Medication characteristics and doses | | | | |
| 3.1 | Varying preferences about characteristics of calcium tablets: Positive perceptions about the characteristics of the calcium tablet played a role in motivating women to take it. Some women preferred the chewable, sweet-tasting tablets that could be swallowed without water, while others preferred the hard tablets which were smaller in size, had no smell and needed to be taken with water. Based on individual preference, the taste, smell, size and convenience were instrumental in uptake of the calcium supplements | 11 27 | Moderate confidence | Due to no or very minor concerns on coherence, minor concerns on methodological limitations (reflexivity), moderate concerns on relevance (all low or lower-middle-income countries and small number of countries) and moderate concerns on adequacy (2 out of 6 studies contributed with moderate to thick data) |
| 3.2 | Medication dosing as a barrier of use: Women described that they could feel overwhelmed with the number of calcium pills they had to take per day, this includes women with comorbidities who need to take additional medications to manage their health condition. Women felt that 3–4 pills per day at multiple times was overly onerous and recommended combining them into one pill could ease the burden | 11 13 27 | High confidence | No or very minor concerns on coherence, minor concerns on methodological limitations (reflexivity), minor concerns on adequacy (3 out of 6 studies contributed with moderate to thick data), and moderate concerns on relevance (small number of countries) |
| 4 | Routine and food insecurity | | | |
| 4.1 | Adherence challenges due to routines: Adherence to calcium consumption was challenging for some women because of conflicting activities in women's daily routine such as consuming other medications, travelling, being away from home and household chores, which make women forgetting to take their medication | 11 13 27 | High confidence | Due to no or very minor concerns with coherence, minor concerns on methodological limitations (reflexivity), moderate concerns on relevance (small number of countries) and moderate concerns on adequacy (small number of studies contributing to qualitative evidence synthesis) |
| 4.2 | Food insecurity as a barrier to calcium uptake: Women believed that adequate foo d was necessary to be able to take the supplements, to avoid nausea associated with the supplement and perceived it as a standard practice to eat before consuming any medication. However, women reported that food insecurity was a critical barrier to calcium uptake | 13 27 | Moderate confidence | Due to no or very minor concern with coherence, minor concerns on methodological limitations (reflexivity), moderate concerns on adequacy (2 out 6 studies contributed with 1 thick and 1 thin data) and moderate concerns on relevance (small number of countries) |
| Strategies to improve use | | | | |

**Table 2** Continued

| No | Summary of qualitative review findings | Contributing qualitative studies | Overall CERQual assessment | Explanation of overall assessment |
|---|---|---|---|---|
| 5.1 | Implementation of reminders to promote adherence: Women and health workers perceived reminders as beneficial in promoting women's adherence in consuming calcium. Several reminder strategies were deemed to be useful by women and health workers, such as home-based posters, calendars with illustrations and daily reminders, and integrating supplement consumption) into the women's daily routine such as at mealtime | 11 13 26 27 | High confidence | Due to no or very minor concerns on methodological limitations, no or very minor concerns on coherence, minor concerns on adequacy (small number of studies contributing to qualitative evidence synthesis), moderate concerns on relevance (due to all studies coming from low-income or lower-middle-income country and small number of countries) |
| 5.2 | Importance of family support and adherence partner implementation: Having family support was instrumental to women in adhering to calcium use and could be leveraged by notifying them on the importance of women's adherence to consumption and appointing someone to be woman's 'adherence partner' to help remind woman in taking the supplement. Both women and health workers were positive about adherence partners in providing support in terms of encouraging them to take the supplements, providing food, helping them around the house, emotional support, improving family relationships, and increased partner or husband involvement in pregnancy | 11 13 26 28 | High confidence | No or very minor concerns on methodological limitations, no or very minor concerns on coherence, minor concerns on adequacy (small number of studies contributing to qualitative evidence synthesis), and moderate concerns on relevance (due to all studies coming from low income or lower-middle-income country only, 2 of 5 studies has indirectly relevant aim, and small number of countries) |
| 5.3 | Counselling facilitates calcium uptake: Both women and health workers and acknowledged that the counselling they received from health workers was a motivator to calcium uptake. Women valued the discussion they have with health workers and felt more confident to take calcium supplements when they received counselling on information of pre-eclampsia/eclampsia and the benefits of calcium from their health workers | 13 27 | Moderate confidence | No or very minor concerns on methodological limitations, no or very minor concerns on coherence, moderate concerns on relevance (due to all studies coming from low or lower-middle-income country and small number of countries) and moderate concerns on adequacy (2 out 6 studies contributed with thick data) |
| **Knowledge and training** | | | | |
| 6.1 | Varied knowledge about pre-eclampsia or eclampsia among health workers: Health workers' knowledge about pre-eclampsia and eclampsia was variable among health workers, and some health workers felt that pre-eclampsia or eclampsia is not a priority health concern in their area and reported never having encountered any case | 26 27 30 | Moderate confidence | Due to no or very minor concerns on methodological limitations and coherence, moderate concerns on relevance (due to all studies coming from low-income or lower-middle-income countries and small number of studies) and moderate concerns on adequacy (2 out 6 studies contributed with thick data) |
| 6.2 | Inadequate training to diagnose and treat pre-eclampsia and eclampsia: While some health workers mentioned that training about pre-eclampsia/eclampsia and calcium supplementation was adequate, others reported that their training lacked depth and continuity, and thus felt unprepared to diagnose and offer information about pre-eclampsia/eclampsia and calcium to pregnant women. Health workers expressed the needs to have more and continuous training to manage the condition and to address any concerns and resistance from the community | 13 26 27 | High confidence | No or very minor concerns on methodological limitations, coherence, moderate concerns on relevance (due to all studies coming from low income or lower-middle-income country and small number of countries) and minor concerns on adequacy (3 out of 6 studies contributed with 2 thick and 1 thin data) |
| **Beliefs on the intervention** | | | | |

Continued

**Table 2** Continued

| No | Summary of qualitative review findings | Contributing qualitative studies | Overall CERQual assessment | Explanation of overall assessment |
|---|---|---|---|---|
| 7.1 | Perceived overmedicalisation when prescribing calcium supplementation: Both health workers and women perceived that prescribing more pills to general low-risk women during pregnancy was a form of overmedicalisation of pregnancy. Some health workers, however, felt that calcium supplementation during pregnancy was a way to prevent further medicalisation when complications occurred | 34 | Low confidence | No or very minor concerns on coherence, minor concerns on methodological limitations (reflexivity and ethics), serious concerns on relevance (evidence coming from high income country only), serious concerns on adequacy (1 out of 6 study contributed) |
| 7.2 | Beliefs that pre-eclampsia is not a serious problem in their settings: Health workers generally had positive beliefs about calcium supplementation and there was optimism that calcium could be delivered through antenatal care health workers. Some facilitators motivating health workers in prescribing calcium supplementation include belief in its prevention value and expected benefits, women liked the calcium supplements, and benefits experienced by women, and perceived lack of knowledge on how to treat pre-eclampsia which motivated health workers to side towards prevention | 13 27 34 | High confidence | No or very minor concerns on coherence, minor concerns on relevance (small number of countries), minor concerns on methodological limitations (ethics and reflexivity), and minor concerns on adequacy (3 out of 6 studies contributed with 2 thick and 1 thin data) |
| Structural factors | | | | |
| 8.1 | Increased workload In general, health workers felt that their workload increased by including calcium supplementation in the services they were providing to pregnant women. Health workers reported inadequate number of staff providing care, yet they needed to provide additional counselling and prescribing to women, especially pregnant women with comorbidities. | 13 27 | Moderate confidence | No or very minor concerns on methodological limitations, no or very minor concerns on coherence, minor concerns on relevance (due to all studies coming from low income or lower-middle-income countries and small number of countries) and serious concerns on adequacy (2 out 6 studies contributed) |

CERQual, Confidence in the Evidence from Reviews of Qualitative research.

One paper, evaluating calcium supplement intake before and after nutritional interventions, showed an association between maternal knowledge of nutrition and higher calcium supplement intake) ($\beta$=31.9, 95% CI 20.9 to 43.0), however, the paper also highlighted there were still large gaps between knowledge and practices, as the intake of calcium supplement tablets during 6 months was low, 82±66 out of the recommended 180 tablets.[34]

### Women's beliefs about calcium supplements in pregnancy
#### Fears about side effects as a barrier to calcium supplements uptake
Women's fears about the side effects of calcium supplements affected their adherence. Women highlighted that assurance of safe use of calcium supplement is a key facilitator to consistent use. However, some women felt safety was not assured by healthcare providers, especially when calcium supplements were perceived as 'experimental'. Women had also received messages from their families or communities that any pills consumed during pregnancy could be harmful (Finding 2.1—High confidence).[18 19 39 40] Quantitative evidence supported qualitative findings as few women reported being discouraged to take supplements by friends (4%) and elder women (3%).[28]

### Women's experiences of side effects
Some women reported experiencing side effects after taking calcium and iron-folic acid supplements, such as dizziness, vomiting, nausea, stomach aches, loss of appetite, tiredness, diarrhoea, bloating and burping, yet noted that side effects subsided with time. Women also reported that they continued taking calcium supplements despite these side effects (Finding 2.2—High confidence).[19 39 40] Quantitative evidence supported the qualitative findings as some women reported experiencing side effects, usually related to gastrointestinal symptoms.[29–31 37] The reported side effect rates were usually low, 4% of women mentioning nausea or vomiting or constipation in one paper while in another paper 14.9% of women reported that side effects were the reason for missing a dose of IFA or calcium.[30 37] These could cause supplement discontinuation or erratic supplement intake for 1–10 days after side effects were felt in around 5% of women.[37]

### Concerns about being stigmatised as HIV patients

Women expressed concerns that if they took calcium supplements, they could be stigmatised as HIV patients, which was a reported barrier to use. Some women were afraid of being stigmatised as their community often associated nutritional supplement intake and accompanying reminder posters, with HIV (Finding 2.3—Moderate confidence).[18 39]

### Positive perceptions of calcium supplements

Women reported that both their perceptions about expected benefits and previous experiences of taking calcium and iron-folic acid supplements were facilitators of use. Women believed that taking pills could compensate for suboptimal nutrition during pregnancy, and that supplements during pregnancy would help keep their baby safe (Finding 2.4—High confidence).[18 19 39–41] Women also reported reduced cravings to consume soil (pica) during pregnancy (a cultural practice or cravings characterised by recurrent ingestion of unusually high amounts of soil, and is related to iron deficiency anaemia, which is common during pregnancy).[39] Furthermore, some women appreciated the emphasis of 'prevention' when encouraging supplement intake.[19 41] Quantitative evidence supported the qualitative findings that women's beliefs about the importance of calcium supplements to both woman's and baby's health are associated with calcium supplement intake.[32] Positive beliefs about calcium supplements and self-efficacy were associated with taking calcium supplements (OR 4.6, 95% CI 2.0 to 10.5) and with taking a higher number of supplements (OR 2.77, 95% CI 1.68 to 4.57).[32]

### Calcium supplement characteristics and regimens
#### Varying preferences about characteristics of calcium tablets

Positive perceptions about the characteristics of the calcium tablet played a role in motivating women to take it. Some women preferred the chewable, sweet-tasting tablets that could be swallowed without water, while others preferred the hard tablets which were smaller in size, had no smell, and needed to be taken with water. Based on individual preference, the taste, smell, size and convenience affected calcium supplement use (Finding 3.1—Moderate confidence).[19 40]

Quantitative evidence supported qualitative findings regarding varying preferences calcium supplement's organoleptic properties. One paper that evaluated the impact of a programme to implement calcium supplementation showed that most women (77%) reported preferences for conventional tablets that were easier to take and swallow, while the least preferred vehicle was unflavoured powder, as women dislike the taste.[30] Another paper reported that conventional tablets had an acceptable taste (83.9% of women).[31] Chewable tablets were preferred by most women (74%) in another paper.[40] Some characteristics that women considered while taking the supplements include tablet's flavour, chewable or swallow, taken with water or not, smell and size.[30 31 40]

### Supplement regimen as a barrier to use

Women described that they feel overwhelmed with the number of calcium tablets they had to take each day, especially women with other comorbidities who needed to take additional medications for their health conditions. Women felt that 3–4 pills per day at multiple times was onerous and preferred if they were combined into one pill (Finding 3.2—High confidence).[19 39 40] In two quantitative studies, women preferred taking fewer tablets per day.[30 40] However, those allocated to the study arm that used more frequent doses took more calcium overall.[40]

### Daily routines and food insecurity
#### Adherence challenges due to routines

Adherence to calcium supplements was challenging for some women because of conflicting activities in their daily routine, such as taking other medications or supplements, travelling, being away from home, and household chores, which can lead them to forget to take calcium (Finding 4.1—High confidence).[19 39 40] In quantitative evidence, women described busy work schedules and not being at home as also contributing to forgetting to consume their calcium supplements.[30 31 37] Forgetting to take calcium supplements was the most frequent reason (52.1%) for not taking IFA or calcium supplements and busy work schedules were inversely associated with adherence to calcium supplementation.[31 37]

#### Food insecurity as a barrier to calcium uptake

Women believed that adequate food was necessary to take the supplements and to avoid nausea. They perceived it as normal to eat before consuming any medication. However, women reported that food insecurity was a critical barrier to calcium uptake (Finding 4.2—Moderate confidence).[19 39] Quantitative evidence supported qualitative findings that women with food security, high socioeconomic status and living in urban areas are more likely to consume calcium supplements as compared with their counterparts.[34 36] Food security was associated with taking six more calcium tablets.[34]

### Strategies to improve the use of calcium supplements
#### Implementation of reminders to promote adherence

Women and healthcare providers perceived reminders as beneficial in promoting women's adherence to calcium supplements. Several reminder strategies were described as useful by women and healthcare providers, such as home-based posters, calendars with illustrations and daily reminders and integrating calcium supplement intake into women's daily routine, such as mealtimes (Finding 5.1—High confidence).[19 38–40] Quantitative evidence supported the qualitative findings that distribution of behaviour change materials to women, such as pill-taking calendars, were associated with increased adherence.[40]

#### Importance of family support and 'adherence partner' implementation

Having family support was instrumental to pregnant women adhering to calcium supplements. This could

be leveraged by notifying family members on the importance of calcium, and appointing someone to be an 'adherence partner' or 'pill buddy' to help remind her to take it. Both women and healthcare providers were positive about adherence partners in providing support in terms of encouraging them to take the supplements, providing food, helping them around the house, providing emotional support, improving family relationships and thereby increasing partner or husband involvement in pregnancy (Finding 5.2—High confidence).[19 38–41] Women could choose who their adherence partner was, and some opted for their husband, a male or female relative, or their child. Some women reported that the support they received from adherence partners decreased over time,[19 39–41] suggesting challenges with sustaining appropriate intake throughout pregnancy.

Quantitative evidence supported the qualitative findings that social support is important in encouraging women to take calcium supplements.[28 32 34 35 40] Women considered involving a husband, partner or family in education sessions or appointing someone as the 'adherence partner' to be an acceptable strategy for promoting adherence.[28 32 34 35] Women often chose their husband (52%), older female relative (23%), children (14%), cousins or other relatives as adherence partners (8%),[28] and were satisfied with the reminders and support received from their adherence partner.[28 32 34] However, a randomised trial assessing adherence partners in improving calcium supplement intake showed that high social support, instead of adherence partners alone, was associated with higher adherence to calcium supplements (OR 2.10; 95% CI 1.32 to 3.34).[28] Women with high family support reported higher intake of calcium supplements (OR=2.1).[32]

### Counselling facilitates calcium supplements uptake

Both women and healthcare providers acknowledged that counselling women on the benefits of calcium was a motivator to calcium supplement intake. Women valued the discussion they have with healthcare providers and felt more confident to take calcium supplements when they received counselling and information and pre-eclampsia and the benefits of calcium from their healthcare providers (Finding 5.3—Moderate confidence).[19 39] This was confirmed by healthcare providers who reported that they have seen positive results following counselling women on iron-folic acid supplements and believed that this would be replicated for calcium-containing supplements.[39]

Quantitative evidence extended qualitative findings where not only counselling, but also starting antenatal contacts at early gestational age, higher number of antenatal contacts, and receiving free calcium supplements were associated with higher calcium intake by women.[31 32 34 36] One paper reports that women were 59 times more likely to consume calcium supplements if they had received them for free.[32]

## Healthcare provider factors
### Healthcare provider knowledge and training
*Varied knowledge about pre-eclampsia among healthcare providers*

Healthcare providers' knowledge about pre-eclampsia was varied. Some felt that pre-eclampsia is not a priority health concern in their area and reported never having encountered any case (Finding 6.1—Moderate confidence).[19 38 42]

*Inadequate training to diagnose and treat pre-eclampsia*

While some healthcare providers mentioned that training about pre-eclampsia and calcium supplements was adequate, others reported that their training lacked depth and continuity, and thus, felt unprepared to diagnose it and offer information these conditions and the use of calcium for prevention. Healthcare providers expressed the need to have more and continuous training on managing pre-eclampsia, as well as time to address concerns or resistance from the community (Finding 6.2—High confidence).[19 38 39] Healthcare providers had positive views of trainings and IEC materials and felt that it helped improve their knowledge.[42] They also valued supervision visits which heled them solve problems and increases their accountability.[42] Support was also needed throughout calcium roll-out to ensure challenges can be addressed promptly during implementation.[19]

### Beliefs about the intervention
*Perceived overmedicalisation when prescribing calcium supplements*

Both healthcare providers and women perceived that prescribing more tablets to 'low-risk' women during pregnancy was a form of overmedicalisation of pregnancy. However, some healthcare providers felt that calcium supplements were a way to prevent further medicalisation due to pre-eclampsia-related complications (Finding 7.1—Low confidence).[18]

*Beliefs about calcium supplements*

Healthcare providers generally had positive beliefs about calcium supplements and there was optimism that calcium could be delivered through antenatal care healthcare providers. Some facilitators motivating healthcare providers to prescribe calcium supplements included their beliefs in its prevention value and expected benefits, that women liked the calcium supplements and experienced benefits from it, and a perceived lack of knowledge on how to treat pre-eclampsia which motivated healthcare providers to side towards prevention (Finding 7.2—High confidence).[18 19 39]

### Structural factors
*High workload, inadequate staffing, stock out and lack of equipment*

In general, healthcare providers felt that their workload increased by including calcium supplements in antenatal care for pregnant women. Healthcare providers reported existing inadequate staffing, yet they needed to provide

additional counselling and prescription to women, especially pregnant women with comorbidities (Finding 8.1—Moderate confidence).[19 39] Stock-outs were also reported as critical barriers, often due to logistical issues in the supply chain (eg, centralisation or procurement changes), errors in demand estimation by government staff and inadequate storage facilities.[42] Importantly, some healthcare providers reported that a lack of equipment to diagnose pre-eclampsia was a barrier to calcium implementation.[19] Facility staff members and supervisors reported that utilisation of health information systems to monitor calcium stocks, checklists to provide feedback on counselling and gaps to address, as well as collaboration with government staff members, could be facilitators of use.[42]

Quantitative evidence extended qualitative findings by showing that in the context of a comprehensive integrated programme including the implementation of job aids, training, guidelines, monitoring and feedback session for healthcare providers, could overcome barriers in prescribing women with calcium supplements.[33]

### Mapping to behaviour change models

Through COM-B and TDF mapping (online supplemental appendix 7), we identified that the critical domains on facilitators and barriers to improve calcium use among women, which include: knowledge, beliefs about consequences, beliefs about capabilities, emotion, social influences, and environmental context and resources to improve calcium use by women. To encourage calcium prescription by healthcare providers, facilitators and barriers related to knowledge, skills, beliefs about consequences and environmental context domains should be addressed. Figure 2 shows the categorisation of barriers and facilitators across the COM-B. The mapping shows that factors encouraging women's use of and adherence to calcium includes receiving adequate information about pre-eclampsia through counselling with healthcare providers and IEC materials, assurance of calcium safety, receiving preferred characteristics of tablet and doses, family and community support, early and frequent antenatal contacts, free calcium supplements, and reminder tools distribution. Likewise, factors that may encourage healthcare providers to prescribe calcium supplements include continuous training about identification and management of pre-eclampsia and calcium supplements, dissemination of consistent messages, reminders and ensuring adequate number of human resources, equipment for diagnosing pre-eclampsia and availability of calcium pills at health facilities.

### DISCUSSION

We included 18 papers from 16 studies conducted primarily in LMICs and reporting views of women, adherence partners and healthcare providers. Our review shows the importance of healthcare providers' knowledge and training about calcium supplements and pre-eclampsia, as women reported providers as the most reliable sources of information and reassurance on safety of calcium and would encourage adherence. Promoting early initiation of

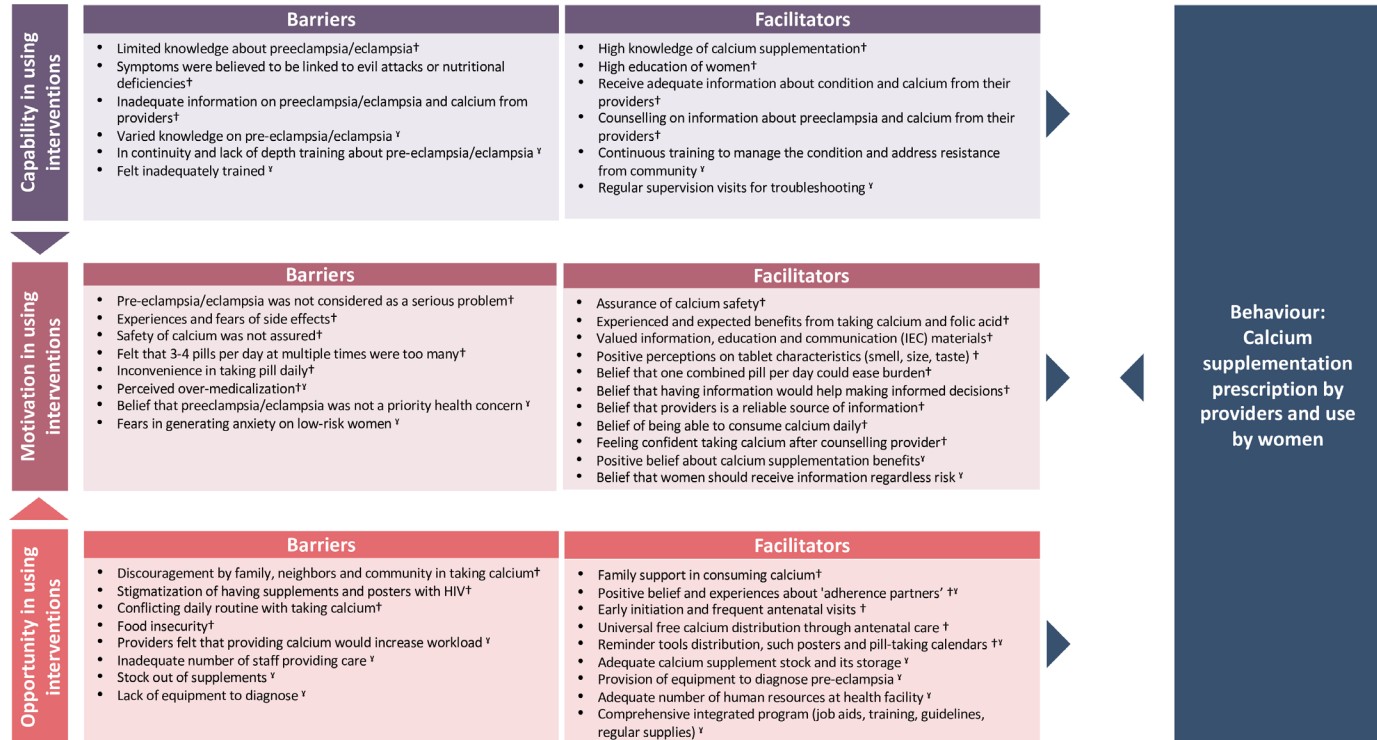

**Figure 2** Capability, Opportunity and Motivation Model of Behaviour categorisation of barriers and facilitators across the Capability, Opportunity and Motivation model of Behaviour. †Woman's factors; ᵛproviders factors.

antenatal care and consistent messages on pre-eclampsia and calcium supplementation may improve women's use of calcium supplements. Free calcium supplements and options on doses and tablet preferences could help overcome barriers to calcium supplement use for women. Reminder systems and support from family and community may also help increase women's calcium uptake.

Women play an important role in decision-making about calcium supplement during pregnancy. Our review shows that women's limited knowledge about and fears of side effects, and potential impacts on their baby serve as the critical barriers to calcium supplement intake. These barriers to calcium supplementation have been reported on the use of other supplements during pregnancy. For example, studies on factors affecting multiple micronutrient supplementation, iron folic acid and lipid-based nutrients reported women's knowledge, acceptance, motivation, and attitudes towards the medication play an important role.[43 44] Furthermore, a study on factors affecting use of intervention for preterm management also reported women felt hesitant to consume the medications to improve labour outcome due to fears about baby's growth and development.[45] This highlights the need to ensure that women are aware on the benefits of the supplement and given assurance on the safety of its use.

As the most trusted informants, healthcare provider's knowledge about calcium and pre-eclampsia is important to support women's uptake of calcium supplements. Studies reported different levels of knowledge about calcium and pre-eclampsia among providers, with some providers having persistent beliefs that evidence on calcium supplements during pregnancy is still in the experimental phase. Some healthcare providers might be reluctant to talk about pre-eclampsia with the fear to stress low risk women; however, this reluctance should be balanced by the risk of symptoms of pre-eclampsia going unnoticed by women. Reinforcing messages related to improving pregnancy outcomes and similarities to other supplements taken during pregnancy such as folic acid and iron that also help to may facilitate use of calcium supplements. Healthcare providers also need to know about how to assess eligibility for calcium supplements, including how to screen and score women at high risk and to identify populations with low calcium intake. There is lack of acceptable biomarkers of individual calcium intake and calcium status, which complicates screening individuals.[46] WHO recommendations on calcium supplementation were set for populations with low calcium intake, as dietary assessments are more reliable to identify populations with low calcium intake rather than to identify individuals.[46]

Human resource shortages are a recurrent health system challenge, particularly in LMICs which results in overburdened staff unable to deliver recommended practices. Appropriate staffing, particularly of midwives and nurses who provide most antenatal care services, remains crucial to achieve quality of care. Unfortunately, this is also applicable on the context of other medication delivered during pregnancy. For example, providers reported unavailability of stock, inadequate staffs and equipment as the main barriers in prescribing interventions to pregnant women experiencing preterm labour. Therefore, ensuring availability of diagnostic tools and calcium stocks is critical to ensure appropriate prescription and delivery of care.[45] Where health providers' constraints persist, innovative strategies to streamline antenatal care practices may be needed to improve efficiency.[47]

### Implications for research, policy and practice

The TDF and COM-B mapping in our review can be used by researchers and programme managers to inform the development of implementation models to optimise the use of calcium supplements. Assessing the extent to which the barriers and facilitators to calcium prescription and use identified in our review are potential implementation challenges in different contexts can be a useful starting point for formative research to scale up implementation. Table 3 presents a list of questions derived from our findings and may help programme implementers, policymakers, researchers and other stakeholders to identify and address factors that may affect prescription and use of calcium supplements during pregnancy. Assessing the extent to which the barriers and facilitators identified in our review are potential implementation challenges in different settings is a useful starting point for formative research to scale this intervention.

### Strengths and limitations

Most included studies were from LMICs and almost half came from the same project conducted in Kenya and Ethiopia, which may limit transferability of results to other contexts. The results from our review therefore can be used to guide formative research as well as implementation and programme planning in other contexts. Most included studies engaged women who attended antenatal care; therefore, might not be representative of those that do not reach the health system during pregnancy. None of the included studies reported the perspectives of policy-makers.

Despite these limitations, this is the first systematic review of barriers and facilitators to calcium supplement use during pregnancy. We adopted a mixed-methods approach which allowed for inclusion and consolidation of studies with a range of designs. Mapping the review findings to behaviour change models improved understanding of how barriers and facilitators influence calcium uptake, and consequently can be addressed in future interventional or programmatic work.

### CONCLUSION

Our review identified a range of barriers and facilitators affecting calcium supplements during pregnancy to prevent pre-eclampsia. When formulating intervention and policies on calcium supplement use, relevant

**Table 3** Implications for research, policy and practice

| Domain | List of questions |
|---|---|
| Guidelines and protocols | Are guidelines and clinical protocols on pre-eclampsia/eclampsia and calcium supplements during pregnancy consistent between WHO, national and facility-levels? |
| Knowledge and learning | 1. Do women or healthcare providers have scepticism or concerns about adverse effects of calcium supplements during pregnancy that can be addressed?<br>2. Do women and their family members receive education and educational materials about signs of pre-eclampsia/eclampsia early in pregnancy?<br>3. Do women have sufficient time and opportunity to discuss pre-eclampsia/eclampsia with healthcare providers during antenatal care?<br>4. Do women have sufficient time and opportunity to discuss calcium supplementation with healthcare providers during antenatal care, including addressing fears about side effects, managing side effects, safety concerns and reinforcing positive messaging about expected benefits?<br>5. Have concerns from both women and healthcare providers about calcium supplements as a form of overmedicalisation of pregnancy been addressed in culturally appropriate ways?<br>6. Have healthcare providers received in-service training on pre-eclampsia/eclampsia prevention and management, including the importance of calcium supplements during pregnancy for prevention? |
| Strategies to improve use | 1. Do all relevant cadres of healthcare providers (including midwives and nurses) have authority to prescribe calcium supplements during pregnancy?<br>2. Do women have the opportunity to try different types of calcium tablets to suit their preferences, such as chewable/non-chewable, different tastes and different size tablets?<br>3. Do women have the opportunity to try different calcium dosing combinations to suit their schedules and preferences?<br>4. For women experiencing or at risk of food insecurity during pregnancy, are there additional social services to support adequate nutrition intake during pregnancy?<br>5. Have different types of reminder systems (eg, posters for home, calendars and integration into daily routines) been designed with women and their families to encourage use?<br>6. Has support from family and/or adherence partners been integrated for women, and do family members or adherence partners have the opportunity to attend educational sessions?<br>7. Are stocks of calcium readily available in the antenatal care wards?<br>8. Is there sufficient funding and budget allocation to ensure continuous procurement and distribution of calcium supplements? |

stakeholders should consider the identified barriers and facilitators to optimise uptake. Findings from this study can inform implementation considerations, to ensure effective and equitable provision and scale-up of calcium interventions.

**Author affiliations**
[1]Centro de Investigaciones en Epidemiología y Salud Pública (Consejo Nacional de Investigaciones Científicas y Técnicas- CONICET), Instituto de Efectividad Clinica y Sanitaria, Buenos Aires, Argentina
[2]Universidad Nacional de La Matanza, San Justo, Provincia de Buenos Aires, Argentina
[3]University of Nairobi, Nairobi, Kenya
[4]Gender and Women's Health Unit, Nossal Institute for Global Health, Melbourne School of Population and Global Health, The University of Melbourne, Melbourne, Victoria, Australia
[5]WHO Collaborating Centre for Global Women's Health, Institute of Metabolism and Systems Research, University of Birmingham, Birmingham, UK
[6]NIHR Birmingham Biomedical Centre (BRC), University Hospitals Birmingham, Birmingham, UK
[7]Nutrition and Food Safety, World Health Organization, Geneve, Switzerland
[8]Effective Care Research Unit, University of the Witwatersrand Faculty of Health Sciences, East London, Eastern Cape, South Africa
[9]University of Botswana, Gaborone, Botswana
[10]Warwick Clinical Trials Unit, University of Warwick, Coventry, UK
[11]Department of Epidemiology, Maastricht University, Maastricht, Netherlands
[12]Maternal, Child and Adolescent Health Program, Burnet Institute, Melbourne, Victoria, Australia
[13]Institute for Clinical Effectiveness and Health Policy, Buenos Aires, Argentina
[14]Centre for Health Economics, University of York, York, UK
[15]Centro de Estudios de Estado y Sociedad, Buenos Aires, Argentina
[16]ACTION ON PREECLAMPSIA, Accra, Ghana
[17]Centro Rosarino de Estudios Perinatales (CREP), Rosario, Santa Fe, Argentina
[18]Keele University, Keele, UK
[19]Human Reproduction Program/World Health Organization (Geneva), Geneva, Switzerland
[20]Centre for Evidence-based Health Care, Division Epidemiology and Biostatistics, Dept. of Global Health, Stellenbosch University, Cape Town, South Africa
[21]UNDP-UNFPA-UNICEF-WHO-World Bank Special Programme of Research, Development and Research Training in Human Reproduction (HRP), Department of Sexual and Reproductive Health and Research, World Health Organization, Geneve, Switzerland
[22]Birmingham Women's and Children's NHS Foundation Trust, Birmingham, UK

**Contributors** All authors have made substantial contributions to the following: Conceptualisation: ST, APB, MB, JA and GCormick. Data curation: TR, JA, GCormick, HM and RIZ. Formal analysis: RIZ, GCormick, HM and MB, Funding acquisition: ST, APB and JA. Investigation and methodology: RIZ, MB, GCormick and HM. Writing–original draft: GCormick, HM, RIZ and MB. Writing–review and editing: GCormick, HM, RIZ, JA, TR, J-PP-R, ZPQ, GJH, HM, LS, JPV, AP, GNG, EA, KKL, GCarroli, RR, KIS, AT, TY, APB, ST and MB. GCormick, HM, RIZ and MB are the guarantors and verified the data in the study. All co-authors accept responsibility for the decision to submit for publication. The corresponding author attests that all listed authors meet authorship criteria and that no others meeting the criteria have been omitted.

**Funding** This work was funded by the Medical Research Council (grant number MR/T038861/1) and the UNDP-UNFPA-UNICEF-WHO-World Bank Special Programme of Research, Development and Research Training in Human Reproduction (HRP), a cosponsored programme executed by the WHO. MB's time is supported by an Australian Research Council Discovery Early Career Researcher Award (DE200100264) and a Dame Kate Campbell Fellowship (University of Melbourne Faculty of Medicine, Dentistry and Health Sciences) and Programa PROINCE-UNLaM-CyTMA SAL-077.

**Competing interests** None declared.

**Patient and public involvement** Patients and/or the public were not involved in the design, or conduct, or reporting, or dissemination plans of this research.

**ORCID iDs**
Gabriela Cormick http://orcid.org/0000-0001-7958-7358
Rana Islamiah Zahroh http://orcid.org/0000-0001-7831-2336
John Allotey http://orcid.org/0000-0003-4134-6246
Thaís Rocha http://orcid.org/0000-0003-0113-6877
Hema Mistry http://orcid.org/0000-0002-5023-1160
Luc Smits http://orcid.org/0000-0003-0785-1345
Joshua Peter Vogel http://orcid.org/0000-0002-3214-7096
Alfredo Palacios http://orcid.org/0000-0001-7684-0880
George N Gwako http://orcid.org/0000-0001-7245-8959
Edgardo Abalos http://orcid.org/0000-0001-6653-429X
Richard Riley http://orcid.org/0000-0001-8699-0735
Ana Pilar Betran http://orcid.org/0000-0002-5631-5883
Meghan A Bohren http://orcid.org/0000-0002-4179-4682

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
