## [Reviewer comments · BMJ Open]

ARTICLE DETAILS

TITLE (PROVISIONAL)	Factors affecting the implementation of calcium supplementation strategies during pregnancy to prevent pre eclampsia: a mixed-methods systematic review
AUTHORS	Cormick, G; Moraa, Helen; Zahroh, Rana; Allotey, John; Rocha, Thaís; Pena-Rosas, Juan-Pablo; Qureshi, Zahida; Hofmeyr, G Justus; Mistry, Hema; Smits, Luc; Vogel, Joshua; Palacios, Alfredo; Gwako, George N.; Abalos, E; Larbi, Koiwah Koi; Carroli, Guillermo; Riley, Richard; Snell, Kym IE; Thorson, Anna; Young, Taryn; Betran, Ana Pilar; Thangaratinam, Shakila; Bohren, Meghan

VERSION 1 – REVIEW

REVIEWER	Robalino, Shannon Oregon Health & Science University School of Medicine, Center for Evidence-based Policy
REVIEW RETURNED	22-Mar-2023

GENERAL COMMENTS	Peer review of: Factors affecting the implementation of calcium supplements strategies during pregnancy to prevent pre-eclampsia: a mixed methods systematic review General comments  1. You use reports and studies interchangeably and this is confusing. Generally researchers refer to studies and publications or papers. A single study might have multiple publications. I suggest removing all of the parentheticals of number of reports as it is unnecessary and confusing. For example, on Page 13 of the PDF, Line 47, Line 50, Line 55. 2. Please clarify which are unique or discrete studies. As an example on Page 13 of the PDF, Line 55-56, you state “One study (7 reports) was conducted in the African Region: Kenya and Ethiopia (19,27-32).”  a. Is this a singular study with 7 different publications/papers?, or b. Is this multiple studies conducted in the same region, by the same author group? 3. If you are speaking about a specific study it would be useful to name that study (e.g., reference 19 Birhanu et al., 2018 is the MICa Trial) 4. Unless it’s a requirement of the journal, there’s no need to also state the title of a table when referring to it in the manuscript. For example, PDF Page 14 Line 35 “...findings (Table 2: Summary of qualitative findings)...” 5. This may be personal preference, but here you use consuming/consume/consumption
--

supplements to me it would make more sense to use using/use as consuming is more related to food and beverages.

6. In the results section, there are many instances where you write “Quantitative evidence extended...” or “Quantitative evidence supported...” or similar. This is somewhat strange phrasing and the link isn’t often clear. Could you elaborate how the quantitative studies supported the qualitative studies? Are there statistics or other measurements than can be used to illustrate how the quantitative studies support the qualitative findings? A good example of where you have done this well is in the last paragraph on PDF Page 20

7. Check order of appendices

8. Be consistent with the use of italics for your in-text confidence ratings

9. Be consistent with the use of acronyms

Search strategy

- Add information about the interface used (e.g., Ovid, ProQuest)
- It’s unclear if an information specialist or medical librarian assisted with this review.

While the search is relatively simple, which seems fine for this topic. However, the use of the Humans limit in the Ovid database requires that individual records are indexed and included the MeSH term Humans. Unfortunately, using this filter eliminates a large number of results published in the last 6 to 18 months due to delays in indexing. The use of the animal terms would have been sufficient to remove the majority of non-human studies. This does mean you likely missed relevant studies though these may have been picked up via your supplemental searches. I’m including a list of 17 citations that may be relevant to your publication.

Table 1

- Remove the number column; it’s unnecessary and confusing
- Remove the Title column; it’s unnecessary
- I suggest reorganizing the table by study name (Project column) and listing the related publications together. Currently the organization by broad study design is confusing and you have more specific information in the Study Designs column.

Table 2

- While I appreciate that the text here is abbreviated and you provide the overall CERQual assessment, the text is identical to that in the manuscript and Appendix 5. I suggest removing the descriptive text under the ‘Summary of qualitative review findings’ column and leaving only the headline themes and subthemes.
- Rename the column ‘Summary of qualitative review findings’ to ‘Themes and Subthemes’

Table 3

- This is a really useful table of questions for program managers and policymakers to consider.

Thank you!

PDF Page 8

- Line 30: “WHO recommendations revalidated”
- Line 42: “facilitators of using calcium”

	PDF Page 9  • Lines 8-9: Should the full name of ENTREQ be appropriately capitalized as you have done for PRISMA? • Line 30: If “capsule” and “capsule filled with liquid” are different, amend to “...chewable table, capsule, liquid filled capsule...” • Line 55: Amend “...retrieved and assessed; disagreements were...” • Line 59: Amend “...other than English, French, or Spanish were...” PDF Page 10  • Line 3: Amend “...French, or Spanish, we would have sought formal...” • Line 56: Spell out GRADE when first introduced PDF Page 11  • Line 3: Amend “...adequacy (25), and relevance (26).” • Lines 16-24: This overview of your results is confusing because some of the information is repeated in a different way on PDF Page 12. I suggest removing the sentences that begin on Line 17 with “Seven reports were...” and ends on Line 22 with “seven separate studies (18,37-42).” PDF Page 13  • Lines 47, 50-51: Remove parentheticals (1 report), (7 reports), etc. Note these parentheticals should be removed throughout the manuscript • Line 57: Remove “in the African Region:” and amend to “One study, Micronutrient InitiativeCornell University Calcium (MiCa) trial, was conducted in Kenya and Ethiopia.” • Line 58: Remove “the Southeast Asia Region:” • Line 57-59: Clarify the sentence that begins “Six studies in...” It is confusing since you say “Bangladesh and India (five reports) ...one study in Bangladesh...”. Something similar to the above example for line 57. PDF Page 14  • Lines 3-4: Remove “Western Pacific Region:” • Line 4: Remove “European Region:” • Line 29: Remove “This included” • Line 34: Amend “... 3) adequacy of...” • Line 34: Amend “Across all themes there...” PDF Page 17  • You should consider referring to Table 2 and/or Appendix 5 when you make parenthetical references. For example, Line 14 “...(1.1 – High confidence) • Line 15: Amend “...Kenya stated that...” • Line 21: It’s strange to say “Quantitative evidence extended the understanding...” It would be clearer to say, for example, “Two cross-sectional studies supported our findings...” This comment is relevant throughout the manuscript where you use the phrasing “Quantitative evidence” • Line 22: Clarify what is being compared. You provide odds ratios, but there is no context as to what is being compared. “higher knowledge of calcium benefits” by whom? Compared with? This comment is relevant throughout the manuscript. • Line 42: Amend “...and make an “uncomplicated pregnancy...”” • Line 44: Amend “...should be a woman’s choice to decide whether to use calcium...” • Line 59: Clairfy. What do you mean by “media”?
--	--

	PDF Page 18  • Line 20: Reference(s) needed for the sentence ending “perceived as ‘experimental’” • Line 32: Is a reference needed for the sentence ending “subsided with time”? PDF Page 19  • Line 21: Reference(s) needed for the sentence ending “to be taken with water” • Line 42: Amend “...was onerous and preferred...” • Line 43-44: Amend “...preferred to take fewer tablets per day...” • Line 57: Italicize “High confidence” for consistency PDF Page 21  • Line 39-43: References needed for the various statements in this sentence PDF Page 22  • Line 17: Italicize “High confidence” for consistency • Line 29: Italicize “Moderate confidence” for consistency • Line 36-38: Amend “...information systems to monitor...counselling, and gaps to...staff members, could be...” • Line 51: Amend “...barriers to improve ... include: knowledge, beliefs...” • Line 53: Amend “...context and resources.” (remove redundant “to improve calcium use by women”) • Line 53: Amend “...calcium prescription...” PDF Page 22-23  • The two paragraphs under the heading “Mapping to behavior change models” are somewhat redundant. I suggest merging them as appropriate. PDF Page 24  • Lines 6-7: “programme managers” instead of “programme implementers”? • Lines 21-25: Most of this description is redundant to the preceding paragraph. I suggest removing it entirely and adding any additional text to the preceding paragraph. References  1. Willemse JPMM, Smits LJM, Braat MME, Meertens LJE, van Montfort P, van Dongen MC, Ellerbrock J, van Dooren IMA, Duvekot EJ, Zwaan IM, Spaanderman MEA, Scheepers HCJ. [Counseling pregnant women on calcium: effects on calcium intake.] J Perinat Med [Internet]. 2022 [cited 2022 Aug 23]; In: Ovid MEDLINE(R) Epub Ahead of Print [Internet]. http://ovidsp.ovid.com/ovidweb.cgi?T=JS&PAGE=reference&D=medp&NEWS=N&AN=35998889 2. Tesfaye B, Sinclair K, Wuehler SE, Moges T, De-Regil LM, Dickin KL. [Applying international guidelines for calcium supplementation to prevent pre-eclampsia: simulation of recommended dosages suggests risk of excess intake in Ethiopia.] Public Health Nutr [Internet]. 2018 [cited 2018 Oct 15];1-11. In: Ovid MEDLINE(R) Epub Ahead of Print [Internet]. http://ovidsp.ovid.com/ovidweb.cgi?T=JS&PAGE=reference&D=medp&NEWS=N&AN=30319089 3. [[Synthesis of evidence and recommendations for the management of calcium supplementation before and during pregnancy for the prevention of preeclampsia and its complications Sintese de evidencias e recomendacoes para o manejo da suplementacao com calcio antes e durante a gravidez para a prevencao da pre-eclampsia e de suas complicacoes].] Sintesis de evidencia y
--	---

	recomendaciones para el manejo de la suplementacion con calcio antes y durante el embarazo para la prevencion de la preeclampsia y sus complicaciones. Rev Panam Salud Publica [Internet]. 2021 [cited 2021];45:e134. In: Ovid MEDLINE(R) PubMed-not-MEDLINE [Internet]. http://ovidsp.ovid.com/ovidweb.cgi?T=JS&PAGE=reference&D=pmnm6&NEWS=N&AN=3473777 1 4. Liu X, Wang X, Tian Y, Yang Z, Lin L, Lin Q, Zhang Z, Li L. [Reduced maternal calcium intake through nutrition and supplementation is associated with adverse conditions for both the women and their infants in a Chinese population.] Medicine (Baltimore) [Internet]. 2017 [cited 2017 May];96(18):e6609. In: Ovid MEDLINE(R) [Internet]. http://ovidsp.ovid.com/ovidweb.cgi?T=JS&PAGE=reference&D=med14&NEWS=N&AN=2847195 6 5. Kant S, Haldar P, Gupta A, Lohiya A. [Serum calcium level among pregnant women and its association with pre-eclampsia and delivery outcomes: A cross-sectional study from North India.] Nepal J Epidemiol [Internet]. 2019 [cited 2019 Dec];9(4):795-803. In: Ovid MEDLINE(R) PubMednot-MEDLINE [Internet]. http://ovidsp.ovid.com/ovidweb.cgi?T=JS&PAGE=reference&D=pmnm4&NEWS=N&AN=3197001 4 6. Anita A, Ramli N. [The Effect of Supplementation of Calcium on Prevention of Pre - Eclampsia in Pregnant Women at Kuta Baro Community Health Center Aceh Besar, Indonesia.] Open Access Maced J Med Sci [Internet]. 2019 [cited 2019 Apr 15];7(7):1129-1132. In: Ovid MEDLINE(R) PubMed-not-MEDLINE [Internet]. Page 2 http://ovidsp.ovid.com/ovidweb.cgi?T=JS&PAGE=reference&D=pmnm4&NEWS=N&AN=3104909 4 8. Chotboon C, Soontrapa S, Buppasiri P, Muktabhant B, Kongwattanakul K, Thinkhamrop J. [Adequacy of calcium intake during pregnancy in a tertiary care center.] Int J Women Health [Internet]. 2018 [cited 2018];10:523-527. In: Ovid MEDLINE(R) PubMed-not-MEDLINE [Internet]. http://ovidsp.ovid.com/ovidweb.cgi?T=JS&PAGE=reference&D=pmnm4&NEWS=N&AN=3025449 3 9. Feldhaus I, LeFevre AE, Rai C, Bhattarai J, Russo D, Rawlins B, Chaudhary P, Thapa K. [Optimizing treatment for the prevention of pre-eclampsia/eclampsia in Nepal: is calcium supplementation during pregnancy cost-effective?.] Cost Eff Resour Alloc [Internet]. 2016 [cited 2016];14:13. In: Ovid MEDLINE(R) PubMed-not-MEDLINE [Internet]. http://ovidsp.ovid.com/ovidweb.cgi?T=JS&PAGE=reference&D=pmnm3&NEWS=N&AN=2803519 3 10. Asemi Z, Samimi M, Siavashani MA, Mazloomi M, Tabassi Z, Karamali M, Jamilian M,
--	--

Esmailzadeh A. [Calcium-Vitamin D Co-supplementation Affects Metabolic Profiles, but not Pregnancy Outcomes, in Healthy Pregnant Women.] *Int J Prev Med* [Internet]. 2016 [cited 2016];7:49. In: Ovid MEDLINE(R) PubMed-not-MEDLINE [Internet]. <http://ovidsp.ovid.com/ovidweb.cgi?T=JS&PAGE=reference&D=pmnm3&NEWS=N&AN=2707688>

7

11. Omotayo MO, Dickin KL, Chapleau GM, Martin SL, Chang C, Mwanga EO, Kung'u JK, Stoltzfus RJ. [Cluster-Randomized Non-Inferiority Trial to Compare Supplement Consumption and Adherence to Different Dosing Regimens for Antenatal Calcium and Iron-Folic Acid Supplementation to Prevent Preeclampsia and Anaemia: Rationale and Design of the Micronutrient Initiative Study.] *J. public health res.* [Internet]. 2015 [cited 2015 Nov 17];4(3):582. In: Ovid MEDLINE(R) PubMed-not-MEDLINE [Internet]. <http://ovidsp.ovid.com/ovidweb.cgi?T=JS&PAGE=reference&D=pmnm3&NEWS=N&AN=2675137>

2

12. Klemm GC, Birhanu Z, Ortolano SE, Kebede Y, Martin SL, Mamo G, Dickin KL. [Integrating Calcium Into Antenatal Iron-Folic Acid Supplementation in Ethiopia: Women's Experiences, Perceptions of Acceptability, and Strategies to Support Calcium Supplement Adherence.] *Glob. health sci. pract.* [Internet]. 2020 [cited 2020 09 30];8(3):413-430. In: Ovid MEDLINE(R) [Internet]. <http://ovidsp.ovid.com/ovidweb.cgi?T=JS&PAGE=reference&D=med18&NEWS=N&AN=3300885>

5

13. Lawrie TA, Betran AP, Singata-Madliki M, Ciganda A, Hofmeyr GJ, Belizan JM, Purnat TD, Manyame S, Parker C, Cormick G. [Participant recruitment and retention in longitudinal preconception randomized trials: lessons learnt from the Calcium And Pre-eclampsia (CAP) trial.] *Trials* [Internet]. 2017 [cited 2017 Oct 26];18(1):500. In: Ovid MEDLINE(R) [Internet]. <http://ovidsp.ovid.com/ovidweb.cgi?T=JS&PAGE=reference&D=med14&NEWS=N&AN=2907391>

6

14. Morris CD, Jacobson SL, Anand R, Ewell MG, Hauth JC, Curet LB, Catalano PM, Sibai BM, Levine RJ. [Nutrient intake and hypertensive disorders of pregnancy: Evidence from a large prospective cohort.] *Am J Obstet Gynecol* [Internet]. 2001 [cited 2001 Mar];184(4):643-51. In: Page 3 Ovid MEDLINE(R) [Internet]. <http://ovidsp.ovid.com/ovidweb.cgi?T=JS&PAGE=reference&D=med4&NEWS=N&AN=11262466>

15. Hauth JC, Ewell MG, Levine RJ, Esterlitz JR, Sibai B, Curet LB, Catalano PM, Morris CD. [Pregnancy outcomes in healthy nulliparas who developed hypertension. Calcium for Preeclampsia Prevention Study Group.] *Obstet Gynecol* [Internet]. 2000 [cited 2000 Jan];95(1):24-8. In: Ovid MEDLINE(R) [Internet]. <http://ovidsp.ovid.com/ovidweb.cgi?T=JS&PAGE=reference&D=med4&NEWS=N&AN=10636496>

	16. Joffe GM, Esterlitz JR, Levine RJ, Clemens JD, Ewell MG, Sibai BM, Catalano PM. [The relationship between abnormal glucose tolerance and hypertensive disorders of pregnancy in healthy nulliparous women. Calcium for Preeclampsia Prevention (CPEP) Study Group.] Am J Obstet Gynecol [Internet]. 1998 [cited 1998 Oct];179(4):1032-7. In: Ovid MEDLINE(R) [Internet]. http://ovidsp.ovid.com/ovidweb.cgi?T=JS&PAGE=reference&D=med4&NEWS=N&AN=9790393 17. Sibai BM, Ewell M, Levine RJ, Klebanoff MA, Esterlitz J, Catalano PM, Goldenberg RL, Joffe G. [Risk factors associated with preeclampsia in healthy nulliparous women. The Calcium for Preeclampsia Prevention (CPEP) Study Group.] Am J Obstet Gynecol [Internet]. 1997 [cited 1997 Nov];177(5):1003-10. In: Ovid MEDLINE(R) [Internet]. http://ovidsp.ovid.com/ovidweb.cgi?T=JS&PAGE=reference&D=med4&NEWS=N&AN=9396883
--	--

REVIEWER	Perichart-Perera, Otilia Instituto Nacional de Perinatología
REVIEW RETURNED	30-Mar-2023

GENERAL COMMENTS	Good quality review and very relevant for clinical practice and for generating new evidence regarding the design and implementation of calcium supplementation programs during pregnancy in low and middle income countries. Data may be used to improve the effectiveness of this intervention in the prevention of preeclampsia and other complications in clinical practice. Methodology is strong. Qualitative results are well described, and discussed. However, quantitative analysis to support qualitative findings is weak. It is clear that the presented themes have been studied mainly in qualitative studies, but the aim of the review is to include both methodologies. Results should be equally described, even though quantitative results may be scarce or weak. Numerical measures and statistical tests are lacking throughout the results section (for the quantitative parts). The discussion should consider other studies or reviews reporting similar experiences with micronutrient/nutrient supplementation around the world. I suggest to contextualize with the experiences regarding multiple micronutrient supplementation and/or folic acid supplementation in low and middle income countries. Similar issues/barriers have been in some studies/reviews that affect adherence or the effectiveness of routine MMS within antenatal care. Not sure if the final table should be in the discussion section; but it may be appropriate considering it includes recommendations. ** COMMENTS Abstract No limitations on the abstract (Prisma 2020 Checklist). Methods Suggestion: include the search keywords used in the review.
--

	Results 1.- Health care providers information is included in the women´s section and vice versa (may be confusing): Page 17 line 14, line 37-46 Page 20, line 20 Page 21 line 6, line 11-15 Table 2. In health providers section, the theme is different from the one stated in the findings. “Adequacy of resources” vs “Structural factors”. It may be confusing. *In Women's knowledge and learning, in 1.2 information from health care providers is included. *In 5.1 and 5.2 also, information from health care providers is included. 2.- When authors are describing that qualitative findings are supported by quantitative analysis, some description of these analyses is needed. Very few of the findings reported an OR or numerical proportion or effect. In the Prisma 2020 Checklist, in “results of individual studies” the authors stated “not applicable”. However, in the Results section they are generally describing associations or differences observed in individual studies. Even though many times an effect size may not be estimated, the type of association, the direction of the association and/or the frequency of X outcome between groups is needed. This is noted mainly in the pages and lines described below: Page 17, line 21-24 May be more clear to include the number of studies to estimate the OR (or the number of reports from X studies) Page 18 Line 3-5 Important to show the effect of this quantitative analysis. Line 7 “increase the probability of reporting...?” “they will report more frequently.... (X% vs X%)?” Page 18, line 24-25; line 36-38 Numbers are necessary; “higher symptoms in those taking supplements vs those that were not taking..?” Compliers vs non-compliers Page 19 Line 43-44 Any proportion to report? Proportion of women who want fewer tablets, for example? Line 57-60 It is a description. May include a proportion or effect Page 20 Line 28 Include numbers (OR? %?); “associated with increased adherence..” need frequency, probability. OR? Line 56-57—how much higher? Numbers? Page 21, line 19 How they were associated? Page 22 Line 43-44 Describe the quantitative evidence Line 53-56 Only include results here. This sentence is more an interpretation/discussion than a result. Table 3 Below table 3, almost all paragraph (line 21-25) is repeated in the paragraph above (main text). Just completed all the information in the main text. Discussion In the Prisma 2020 Checklist, in “Discussion” (23 a: Provide a general interpretation of the results in the context of other evidence) the authors refer “page 18 to interpretation”. However,
--	---

	page 18 is still the results section. Discussion with interpretation of the results starts in page 23. But there is no inclusion of other evidence, only discussion within the evidence selected for the review. Need evidence to contrast and contextualize your results. What has been done before? Are the findings from this review similar/different to other reviews of studies regarding what barriers and facilitators exist among women in achieving adherence to nutrient supplementation? Difficulties in implementation? Health care provider barriers/facilitators? Evidence from multiple micronutrient supplementation (MMS) studies has reported women’s skills, knowledge, motivation, attitudes, among other themes regarding supplementation. Iron and folic acid supplementation studies also have reported motivators/barriers. Any similarities or differences in the barriers/facilitators that have been reported in other nutrient supplementation implementation experiences in LMICs? In antenatal care? Some suggestions of reviews (Garcia-Casal M.N, Maternal and Child Nutrition, 2018). In the second paragraph: The authors mention the need for experience in screening women for high risk of preeclampsia. May be interesting to add reliable and practical options of this screening for LMICs. Also, add some ideas on what experiences exist about assessing low Ca intake? Limitations: Any limitations of the qualitative studies? Considering main results are derived from this study design. Figures I cannot see the title of Figure 2. Appendix 2 In “Appraisal Results” the authors left it blank. Please complete.
--	---

VERSION 1 – AUTHOR RESPONSE

Reviewer 1	
Thank you for submitting this systematic review on the barriers and facilitators to increased calcium supplementation in pregnant women with perspectives from pregnant women, their partners or other social supports, and healthcare workers. Qualitative systematic reviews are difficult, so well done on this one! For more specific comments, please see the attached PDF.	Thank you for the kind comment.
You use reports and studies interchangeably and this is confusing. Generally researchers refer to studies and publications or papers. A single study might have multiple publications. I suggest removing all of the parentheticals of number of reports as it is	Thank you, we have replaced reports to papers.

unnecessary and confusing. For example, on Page 13 of the PDF, Line 47, Line 50, Line 55.	
Please clarify which are unique or discrete studies. As an example on Page 13 of the PDF, Line 55-56, you state “One study (7 reports) was conducted in the African Region: Kenya and Ethiopia (19,27-32).” a. Is this a singular study with 7 different publications/papers?, or is this multiple studies conducted in the same region, by the same author group? If you are speaking about a specific study it would be useful to name that study (e.g., reference 19 Birhanu et al., 2018 is the MICa Trial)	This is one study with 7 papers. We have now included the MICa Trial in the description.
Unless it’s a requirement of the journal, there’s no need to also state the title of a table when referring to it in the manuscript.	we have deleted the title of table 2.
This may be personal preference, but here you use consuming/consume/consumption supplements to me it would make more sense to use using/use as consuming is more related to food and beverages.	Thank you, we feel that in the context of medication use, it is common to refer to it as medication consumption. We have modify the use of consumption for calcium supplement intake or use depeding on the context.
In the results section, there are many instances where you write “Quantitative evidence extended...” or “Quantitative evidence supported...” or similar. This is somewhat strange phrasing and the link isn’t often clear. Could you elaborate how the quantitative studies supported the qualitative studies? Are there statistics or other measurements than can be used to illustrate how the quantitative studies support the qualitative findings? A good example of where you have done this well is in the last paragraph on PDF Page 20	Thank you for your comment. We mapped the quantitative results to qualitative themes (thematically). We used this approach because the outcomes reported in the quantitative studies were heterogenous and no quantitative meta-analysis was possible – this is a similar approach to a similar review (https://journals.plos.org/plosmedicine/article?id=10.1371/journal.pmed.1004074). When we report “quantitative evidence supported qualitative findings”, it means that the results of the quantitative findings are the same, or in the same direction, as qualitative findings. On another hand, “quantitative evidence extended qualitative findings” means that the quantitative findings added a new finding that’s not captured by qualitative findings, but it is still under the same theme. As suggested we have made some changes to illustrate the findings.
Check order of appendices	Thank you. We have checked and confirmed it is on the right order.

Be consistent with the use of italics for your in-text confidence ratings	We have removed all italics. Thank you.
Be consistent with the use of acronyms	Thank you. We have checked and confirmed it is consistent.
Add information about the interface used (e.g., Ovid, ProQuest)	Thank you, we used the databases listed on the manuscript: MEDLINE and EMBASE via Ovid, CINAHL, Global Health, and grey literature.
It's unclear if an information specialist or medical librarian assisted with this review. While the search is relatively simple, which seems fine for this topic. However, the use of the Humans limit in the Ovid database requires that individual records are indexed and included the MeSH term Humans. Unfortunately, using this filter eliminates a large number of results published in the last 6 to 18 months due to delays in indexing. The use of the animal terms would have been sufficient to remove the majority of non-human studies. This does mean you likely missed relevant studies though these may have been picked up via your supplemental searches. I'm including a list of 17 citations that may be relevant to your publication.	Thank you, we acknowledge the reviewer's comment. The search was performed by an expert librarian and, including consulting with experts on calcium research. A balanced risk decision was made for the final search. We have reviewed the list of 17 paper and only one paper was relevant, however it belongs to an included study and no new information was added to our manuscript. Please find below our comments on each of the 17 citations listed.
Willemse JPMM, Smits LJM, Braat MME, Meertens LJE, van Montfort P, van Dongen MC, Ellerbrock J, van Dooren IMA, Duvekot EJ, Zwaan IM, Spaanderman MEA, Scheepers HCJ. [Counseling pregnant women on calcium: effects on calcium intake.] J Perinat Med [Internet]. 2022 [cited 2022 Aug 23]; In: Ovid MEDLINE(R) Epub Ahead of Print [Internet]. http://ovidsp.ovid.com/ovidweb.cgi?T=JS&PAGE=reference&D=medp&NEWS=N&AN=35998889	Thank you, we have reviewed this study, and this is not eligible to be included in our study as there is no factors influencing calcium use covered/investigated.
Tesfaye B, Sinclair K, Wuehler SE, Moges T, De-Regil LM, Dickin KL. [Applying international guidelines for calcium supplementation to prevent pre-eclampsia: simulation of recommended dosages suggests risk of excess intake in Ethiopia.] Public Health Nutr [Internet]. 2018 [cited 2018 Oct 15];1-11. In: Ovid MEDLINE(R) Epub Ahead of Print [Internet]. http://ovidsp.ovid.com/ovidweb.cgi?T=JS&PAGE=reference&D=medp&NEWS=N&AN=30319089	Thank you, this review has been captured on our search, we have reviewed this study, and this is not eligible to be included in our study as there is no factors influencing calcium use covered/investigated.

[[Synthesis of evidence and recommendations for the management of calcium supplementation before and during pregnancy for the prevention of preeclampsia and its complications Sintese de evidencias e recomendacoes para o manejo da suplementacao com calcio antes e durante a gravidez para a prevencao da pre-eclampsia e de suas complicacoes].] Sintesis de evidencia y recomendaciones para el manejo de la suplementacion con calcio antes y durante el embarazo para la prevencion de la preeclampsia y sus complicaciones. Rev Panam Salud Publica [Internet]. 2021 [cited 2021];45:e134. In: Ovid MEDLINE(R) PubMed-not-MEDLINE [Internet]. http://ovidsp.ovid.com/ovidweb.cgi?T=JS&PAGE=reference&D=pmnm6&NEWS=N&AN=34737771	Thank you, we have reviewed this, and this is not eligible as it is systematic review and not a primary study.
Liu X, Wang X, Tian Y, Yang Z, Lin L, Lin Q, Zhang Z, Li L. [Reduced maternal calcium intake through nutrition and supplementation is associated with adverse conditions for both the women and their infants in a Chinese population.] Medicine (Baltimore) [Internet]. 2017 [cited 2017 May];96(18):e6609. In: Ovid MEDLINE(R) [Internet]. http://ovidsp.ovid.com/ovidweb.cgi?T=JS&PAGE=reference&D=med14&NEWS=N&AN=28471956	Thank you, we have reviewed this study, and this is not eligible to be included in our study as there is no factors influencing calcium use covered/investigated.
Kant S, Haldar P, Gupta A, Lohiya A. [Serum calcium level among pregnant women and its association with pre-eclampsia and delivery outcomes: A cross-sectional study from North India.] Nepal J Epidemiol [Internet]. 2019 [cited 2019 Dec];9(4):795-803. In: Ovid MEDLINE(R) PubMed-not-MEDLINE [Internet]. http://ovidsp.ovid.com/ovidweb.cgi?T=JS&PAGE=reference&D=pmnm4&NEWS=N&AN=31970014	Thank you, this study has been captured on our search, we have reviewed this study, and this is not eligible to be included in our study as there is no factors influencing calcium use covered/investigated.
Anita A, Ramli N. [The Effect of Supplementation of Calcium on Prevention of Pre - Eclampsia in Pregnant Women at Kuta Baro Community Health Center Aceh Besar, Indonesia.] Open Access Maced J Med Sci [Internet]. 2019 [cited 2019 Apr	Thank you, this study has been captured on our search, we have reviewed this study, and this is not eligible to be included in our study as there is no factors influencing calcium use covered/investigated.
Chotboon C, Soontrapa S, Buppasiri P, Muktabhant B, Kongwattanakul K,	Thank you, we have reviewed this study, and this is not eligible to be included in

Thinkhamrop J. [Adequacy of calcium intake during pregnancy in a tertiary care center.] Int J Women Health [Internet]. 2018 [cited 2018];10:523-527. In: Ovid MEDLINE(R) PubMed-not-MEDLINE [Internet]. http://ovidsp.ovid.com/ovidweb.cgi?T=JS&PAGE=reference&D=pmnm4&NEWS=N&AN=30254493	our study as there is no factors influencing calcium use covered/investigated.
Feldhaus I, LeFevre AE, Rai C, Bhattarai J, Russo D, Rawlins B, Chaudhary P, Thapa K. [Optimizing treatment for the prevention of pre-eclampsia/eclampsia in Nepal: is calcium supplementation during pregnancy cost-effective?.] Cost Eff Resour Alloc [Internet]. 2016 [cited 2016];14:13. In: Ovid MEDLINE(R) PubMed-not-MEDLINE [Internet].	Thank you, this study has been captured on our search, we have reviewed this study, and this is not eligible to be included in our study as there is no factors influencing calcium use covered/investigated.
Asemi Z, Samimi M, Siavashani MA, Mazloomi M, Tabassi Z, Karamali M, Jamilian M, Esmailzadeh A. [Calcium-Vitamin D Co-supplementation Affects Metabolic Profiles, but not Pregnancy Outcomes, in Healthy Pregnant Women.] Int J Prev Med [Internet]. 2016 [cited 2016];7:49. In: Ovid MEDLINE(R) PubMed-not-MEDLINE [Internet]. http://ovidsp.ovid.com/ovidweb.cgi?T=JS&PAGE=reference&D=pmnm3&NEWS=N&AN=27076887	Thank you, this study has been captured on our search, we have reviewed this study, and this is not eligible to be included in our study as there is no factors influencing calcium use covered/investigated.
Omotayo MO, Dickin KL, Chapleau GM, Martin SL, Chang C, Mwanga EO, Kung'u JK, Stoltzfus RJ. [Cluster-Randomized Non-Inferiority Trial to Compare Supplement Consumption and Adherence to Different Dosing Regimens for Antenatal Calcium and Iron-Folic Acid Supplementation to Prevent Preeclampsia and Anaemia: Rationale and Design of the Micronutrient Initiative Study.] J. public health res. [Internet]. 2015 [cited 2015 Nov 17];4(3):582. In: Ovid MEDLINE(R) PubMed-not-MEDLINE [Internet]. http://ovidsp.ovid.com/ovidweb.cgi?T=JS&PAGE=reference&D=pmnm3&NEWS=N&AN=26751372	Thank you, this study has been captured on our search, we have reviewed this study and this is not eligible as it is a protocol.
Klemm GC, Birhanu Z, Ortolano SE, Kebede Y, Martin SL, Mamo G, Dickin KL. [Integrating Calcium Into Antenatal Iron-Folic Acid Supplementation in Ethiopia: Women's Experiences, Perceptions of Acceptability, and Strategies to Support Calcium Supplement Adherence.] Glob. health sci. pract. [Internet].	Thank you. This is a second paper from a study included (Martin 2017).

2020 [cited 2020 09 30];8(3):413-430. In: Ovid MEDLINE(R) [Internet]. http://ovidsp.ovid.com/ovidweb.cgi?T=JS&PAGE=reference&D=med18&NEWS=N&AN=33008855	
Lawrie TA, Betran AP, Singata-Madliki M, Ciganda A, Hofmeyr GJ, Belizan JM, Purnat TD, Manyame S, Parker C, Cormick G. [Participant recruitment and retention in longitudinal preconception randomized trials: lessons learnt from the Calcium And Pre-eclampsia (CAP) trial.] Trials [Internet]. 2017 [cited 2017 Oct 26];18(1):500. In: Ovid MEDLINE(R) [Internet]. http://ovidsp.ovid.com/ovidweb.cgi?T=JS&PAGE=reference&D=med14&NEWS=N&AN=29073916	Thank you, this study has been captured on our search, we have reviewed this study, and this is not eligible to be included in our study as there is no factors influencing calcium use covered/investigated.
Morris CD, Jacobson SL, Anand R, Ewell MG, Hauth JC, Curet LB, Catalano PM, Sibai BM, Levine RJ. [Nutrient intake and hypertensive disorders of pregnancy: Evidence from a large prospective cohort.] Am J Obstet Gynecol [Internet]. 2001 [cited 2001 Mar];184(4):643-51. In: Ovid MEDLINE(R) [Internet]. http://ovidsp.ovid.com/ovidweb.cgi?T=JS&PAGE=reference&D=med4&NEWS=N&AN=11262466	Thank you, this study has been captured on our search, we have reviewed this study, and this is not eligible to be included in our study as there is no factors influencing calcium use covered/investigated.
Hauth JC, Ewell MG, Levine RJ, Esterlitz JR, Sibai B, Curet LB, Catalano PM, Morris CD. [Pregnancy outcomes in healthy nulliparas who developed hypertension. Calcium for Preeclampsia Prevention Study Group.] Obstet Gynecol [Internet]. 2000 [cited 2000 Jan];95(1):24-8. In: Ovid MEDLINE(R) [Internet]. http://ovidsp.ovid.com/ovidweb.cgi?T=JS&PAGE=reference&D=med4&NEWS=N&AN=10636496	Thank you, this study has been captured on our search, we have reviewed this study and this is not eligible to be included in our study as there is no factors influencing calcium use covered/investigated.
Joffe GM, Esterlitz JR, Levine RJ, Clemens JD, Ewell MG, Sibai BM, Catalano PM. [The relationship between abnormal glucose tolerance and hypertensive disorders of pregnancy in healthy nulliparous women. Calcium for Preeclampsia Prevention (CPEP) Study Group.] Am J Obstet Gynecol [Internet]. 1998 [cited 1998 Oct];179(4):1032-7. In: Ovid MEDLINE(R) [Internet]. http://ovidsp.ovid.com/ovidweb.cgi?T=JS&PAGE=reference&D=med4&NEWS=N&AN=10636496	Thank you, this study has been captured on our search, we have reviewed this study, and this is not eligible to be included in our study as there is no factors influencing calcium use covered/investigated.

GE=reference&D=med4&NEWS=N&AN=9790393	
Sibai BM, Ewell M, Levine RJ, Klebanoff MA, Esterlitz J, Catalano PM, Goldenberg RL, Joffe G. [Risk factors associated with preeclampsia in healthy nulliparous women. The Calcium for Preeclampsia Prevention (CPEP) Study Group.] Am J Obstet Gynecol [Internet]. 1997 [cited 1997 Nov];177(5):1003-10. In: Ovid MEDLINE(R) [Internet]. http://ovidsp.ovid.com/ovidweb.cgi?T=JS&PA GE=reference&D=med4&NEWS=N&AN=9396883	Thank you, this study has been captured on our search, we have reviewed this study, and this is not eligible to be included in our study as there is no factors influencing calcium use covered/investigated.
Table 1  • Remove the number column; it's unnecessary and confusing • Remove the Title column; it's unnecessary • I suggest reorganizing the table by study name (Project column) and listing the related publications together. Currently the organization by broad study design is confusing and you have more specific information in the Study Designs column. 	Thank you. We have removed numbers, title column, and reorganised the table as suggested.
Table 2 While I appreciate that the text here is abbreviated and you provide the overall CERQual assessment, the text is identical to that in the manuscript and Appendix 5. I suggest removing the descriptive text under the 'Summary of qualitative review findings' column and leaving only the headline themes and subthemes. Rename the column 'Summary of qualitative review findings' to 'Themes and Subthemes'	Thank you for this feedback, however, we created this table following the suggestion and template from GRADE-CERQual coordinating group, as it is easier to see the results and confidence in one place. We have retained the table as it is with no change. Please see the recommendations below: Lewin S, Bohren M, Rashidian A, Munthe-Kaas H, Glenton C, Colvin CJ, Garside R, Noyes J, Booth A, Tunçalp Ö, Wainwright M, Flottorp S, Tucker JD, Carlsen B. Applying GRADE-CERQual to qualitative evidence synthesis findings- paper 2: how to make an overall CERQual assessment of confidence and create a Summary of Qualitative Findings table. Implement Sci. 2018 Jan 25;13(Suppl 1):10.
Table 3	Thank you.

This is a really useful table of questions for program managers and policymakers to consider. Thank you!	
PDF Page 8 Line 30: "WHO recommendations revalidated"	Thank you. WHO normally uses the term "update" to reflect the recommendation change. Thus, we have used the "updated" term instead.
Line 42: "facilitators of using calcium"	We agree, thank you.
PDF Page 9 Lines 8-9: Should the full name of ENTREQ be appropriately capitalized as you have done for PRISMA?	We agree, thank you.
Line 30: If "capsule" and "capsule filled with liquid" are different, amend to "...chewable table, capsule, liquid filled capsule..."	Changed as suggested, thank you.
Line 55: Amend "...retrieved and assessed; disagreements were..."	Changed as suggested, thank you.
Line 59: Amend "...other than English, French, or Spanish were..."	Changed as suggested, thank you.
PDF Page 10 Line 3: Amend "...French, or Spanish, we would have sought formal..."	Changed as suggested, thank you.
Line 56: Spell out GRADE when first introduced	We have added Grading of Recommendations, Assessment, Development, and Evaluations. Thank you
PDF Page 11 Line 3: Amend "...adequacy (25), and relevance (26)."	Amended, thank you.
Lines 16-24: This overview of your results is confusing because some of the information is repeated in a different way on PDF Page 12. I suggest removing the sentences that begin on Line 17 with "Seven reports were..." and ends on Line 22 with "seven separate studies (18,37-42)."	We have deleted these sentences. Thank you
PDF Page 13 Lines 47, 50-51: Remove parentheticals (1 report), (7 reports), etc. Note these parentheticals should be removed throughout the manuscript	We feel it is important to retain this parenthesis to help readers differentiate between study and papers. As it may confuse readers to see 4 studies but there are 7 citations at the end. So, we have retained it as it is. Thank you

Line 57: Remove “in the African Region:” and amend to “One study, Micronutrient Initiative-Cornell University Calcium (MICa) trial, was conducted in Kenya and Ethiopia.”	Removed, thank you.
Line 58: Remove “the Southeast Asia Region:” _	Removed, thank you.
Line 57-59: Clarify the sentence that begins “Six studies in...” _It is confusing since you say “Bangladesh and India (five reports) ...one study in Bangladesh...”. Something similar to the above example for line 57.	We have changed this phrase to “The studies were conducted in seven different countries across four regions. One study, the MICa Trial, was conducted in Kenya and Ethiopia study (7 papers) (19,28,33,38–41). One study conducted in Bangladesh and India (5 papers) (27,32,34,35,42), one study only in Bangladesh (1 papers) (30), two studies only in India (2 papers) (27,29), and two studies in Nepal (2 papers) (31,37). The remaining two studies were conducted in China (36), and the Netherlands (18).” I hope it is clearer and help readers to differentiate between study and papers.
PDF Page 14	removed, thank you
• _Lines 3-4: Remove “Western Pacific Region:” _	
• _Line 4: Remove “European Region:” _	Removed, thank you.
• _Line 29: Remove “This included” _	Removed, thank you.
• _Line 34: Amend “... _3) adequacy of...” _	Amended. Thank you.
• _Line 34: Amend “Across all themes there...” _	
PDF Page 17	
• _You should consider referring to Table 2 and/or Appendix 5 when you make parenthetical references. For example, Line 14 “...(1.1 – _High confidence)	Thank you, Table 2 has been mentioned earlier under qualitative and quantitative synthesis, in the paragraph right before talking on specific themes, thus we feel it will be redundant to mention it all over again. So instead, we have added “Finding” in front of number like Finding 1.1 instead of 1.1, so it is clear it is the numbering of Finding in Table 2.
_Line 15: Amend “...Kenya stated that...” _	Amended. Thank you.
• _Line 21: It’s strange to say “Quantitative evidence extended the understanding...” _It would be clearer to say, for example, “Two cross-sectional studies supported our findings...” _This comment is relevant	Thank you. However, as we are trying to compare qualitative and quantitative findings (whether similar or different barriers and facilitators are identified), we feel it is less important what the

throughout the manuscript where you use the phrasing “Quantitative evidence” _	quantitative study design is (i.e., RCT, cross sectional etc). We feel it is more important, and less words, to refer it as “quantitative findings” instead of the specific quantitative study design. As we are referring to findings from quantitative study, regardless of their study design. No changes made on this.
• _Line 22: Clarify what is being compared. You provide odds ratios, but there is no context as to what is being compared. “higher knowledge of calcium benefits” _by whom? Compared with? This comment is relevant throughout the manuscript.	We have rephrased the sentence. “Quantitative evidence extended the understanding of qualitative findings, where women were more likely to have higher calcium supplement intake if they have higher knowledge of calcium benefits (Odds Ratio (OR) 11.7, 95% Confidence Interval (CI) 5.97-22.86) (32,34) and higher general education (OR 2.59, 95%CI 2.21-3.05) (32,34,36).”
• _Line 42: Amend “...and make an “uncomplicated pregnancy...” _	Amended. Thank you.
• _Line 44: Amend “...should be a woman’s choice to decide whether to use calcium...” _	Amended. Thank you.
• _Line 59: Clairfy. What do you mean by “media”?	The literature refers to mass media campaigns, we added this in the text.
PDF Page 18 • _Line 20: Reference(s) needed for the sentence ending “perceived as ‘experimental’” _	Thank you. We have added the references after the confidence statement, bolded below. This is reported according to the GRADE-CERQual guidance (https://implementationscience.biomedcentral.com/articles/10.1186/s13012-017-0689-2). Women’s fears about the side effects of calcium supplements affected their adherence. Women highlighted that assurance of safe use of calcium supplement is a key facilitator to consistent use. However, some women felt safety was not assured by healthcare providers, especially when calcium supplements were perceived as “experimental”. Women had also received messages from their families or communities that any pills taken during

	pregnancy could be harmful. (Finding 2.1 – High confidence) (18,19,39,40).
• _Line 32: Is a reference needed for the sentence ending “subsided with time”?	We do not need to add ref for each of the sentences as it is summary findings, thus the contributing references are already referred after the confidence assessment, bolded below. This is reported according to the GRADE-CERQual guidance (https://implementationscience.biomedcentral.com/articles/10.1186/s13012-017-0689-2). Some women reported experiencing side effects after taking calcium and iron-folic acid supplements, such as dizziness, vomiting, nausea, stomach aches, loss of appetite, tiredness, diarrhoea, bloating, and burping, yet noted that side effects subsided with time. Women also reported that they continued using calcium despite these side effects (Finding 2.2 – High confidence) (19,39,40).
PDF Page 19 • _Line 21: Reference(s) needed for the sentence ending “to be taken with water” _	We do not need to add ref for each of the sentences as it is summary findings, thus the contributing references are already referred after the confidence assessment, bolded below. This is reported according to the GRADE-CERQual guidance (https://implementationscience.biomedcentral.com/articles/10.1186/s13012-017-0689-2). Positive perceptions about the characteristics of the calcium tablet played a role in motivating women to take it. Some women preferred the chewable, sweet-tasting tablets that could be swallowed without water, while others preferred the hard tablets which were smaller in size, had no smell, and needed to be taken with water. Based on individual preference, the taste, smell, size, and convenience affected calcium supplement use (Finding 3.1 – Moderate confidence) (19,40).

 • _Line 42: Amend "...was onerous and preferred..." _ 	Amended. Thank you.
 • _Line 43-44: Amend "...preferred to take fewer tablets per day..." _ 	Amended. Thank you.
 • _Line 57: Italicize "High confidence" _for consistency 	We have removed italics from all. Thank you.
PDF Page 21  • _Line 39-43: References needed for the various statements in this sentence 	Thank you, as mentioned earlier, we do not need to add refs for each of the sentences on qualitative findings as it is summary findings, thus the contributing references are already referred after the confidence assessment. This is the recommended reporting format for GRADE-CERQual (https://implementationscience.biomedcentral.com/articles/10.1186/s13012-017-0689-2).
PDF Page 22  • _Line 17: Italicize "High confidence" _for consistency 	We have removed italics from all. Thank you.
 • _Line 29: Italicize "Moderate confidence" _for consistency 	We have removed italics from all. Thank you.
 • _Line 36-38: Amend "...information systems to monitor...counselling, and gaps to...staff members, could be..." _ 	Amended. Thank you.
 • _Line 51: Amend "...barriers to improve ... _include: knowledge, beliefs..." _ 	Amended. Thank you.
 • _Line 53: Amend "...context and resources." _ (remove redundant "to improve calcium use by women") 	Amended. Thank you.
 • _Line 53: Amend "...calcium prescription..." _ 	Amended. Thank you.
PDF Page 22-23  • _The two paragraphs under the heading "Mapping to behavior change models" _are somewhat redundant. I suggest merging them as appropriate. 	Thank you. We feel it is important to retain the first paragraph as it is to highlight the critical domains identified from the behavioral change frameworks. We have edited this to be clearer.
PDF Page 24  • _Lines 6-7: "programme managers" _instead of "programme implementers"? 	Thank you, we feel implementers are more appropriate as implementers can be other than programme managers.

• _Lines 21-25: Most of this description is redundant to the preceding paragraph. I suggest removing it entirely and adding any additional text to the preceding paragraph.	Thank you for this suggestion, we have removed it.
Reviewer: 2	
Good quality review and very relevant for clinical practice and for generating new evidence regarding the design and implementation of calcium supplementation programs during pregnancy in low and middle income countries. Data may be used to improve the effectiveness of this intervention in the prevention of preeclampsia and other complications in clinical practice. Methodology is strong. Qualitative results are well described, and discussed. However, quantitative analysis to support qualitative findings is weak. It is clear that the presented themes have been studied mainly in qualitative studies, but the aim of the review is to include both methodologies. Results should be equally described, even though quantitative results may be scarce or weak. Numerical measures and statistical tests are lacking throughout the results section (for the quantitative parts).	Thank you for your kind comment. Our study focuses on the facilitators and barriers affecting the calcium use, thus we are focusing on these reported factors instead of quantification of the results. Furthermore, instead of showing statistical significance from the quantitative studies, the mapping of quantitative findings was done to narratively explained if there are any convergence and divergence on the results reported from qualitative studies, which will make the data on facilitators and barriers are richer. We mapped the quantitative results to qualitative themes (thematically), because the outcomes reported in the quantitative studies were heterogenous and no quantitative meta-analysis was possible – this is a similar approach to a similar review (https://journals.plos.org/plosmedicine/article?id=10.1371/journal.pmed.1004074).
The discussion should consider other studies or reviews reporting similar experiences with micronutrient/nutrient supplementation around the world. I suggest to contextualize with the experiences regarding multiple micronutrient supplementation and/or folic acid supplementation in low and middle income countries. Similar issues/barriers have been in some studies/reviews that affect adherence or the effectiveness of routine MMS within antenatal car	Thank you, we believe it is important to explore the roles of calcium on micronutrients and how it may influence uptake. However, we feel the micronutrient is outside of our scope at this point as we wanted to understand factors influencing use of calcium tablets among pregnant women.
Not sure if the final table should be in the discussion section; but it may be appropriate considering it includes recommendations.	Thank you, yes we feel it fits in the discussion as it is part of an implication.
Abstract No limitations on the abstract (Prisma 2020 Checklist).	Thank you, we have decided to remove this from abstract as it is mentioned on the strength and limitations box near the abstract.

Methods Suggestion: include the search keywords used in the review.	Thank you, we have this already written on the manuscript: A search strategy was developed and adapted for each database (Appendix 3), using different terms for calcium and pregnancy. Detailed keywords can be found on Appendix 3.
Results 1.- Health care providers information is included in the women´s section and vice versa (may be confusing): Page 17 line 14, line 37-46 Page 20, line 20 Page 21 line 6, line 11-15	Thank you, it is because the findings were related to women´s factors, however at times, it does only come from women but also providers. We feel it is important to be clear where the perspectives are coming from.
Table 2. In health providers section, the theme is different from the one stated in the findings. “Adequacy of resources” vs “Structural factors”. It may be confusing.	Thank you. We have unified the theme.
*In Women's knowledge and learning, in 1.2 information from health care providers is included.	Thank you, it is because the findings were related to women´s factors, however at times, it does only come from women but also providers. We feel it is important to be clear where the perspectives are coming from. No changes made.
*In 5.1 and 5.2 also, information from health care providers is included.	Thank you, it is because the findings were related to women´s factors, however at times, it does only come from women but also providers. We feel it is important to be clear where the perspectives are coming from. No changes made.
When authors are describing that qualitative findings are supported by quantitative analysis, some description of these analyses is needed. Very few of the findings reported an OR or numerical proportion or effect.	Thank you, as mentioned earlier, our study focuses on the facilitators and barriers affecting the calcium use, thus we are focusing on these reported factors instead of quantification of the results. The full details of the quantitative results are reported in Appendix 6.

In the Prisma 2020 Checklist, in “results of individual studies” the authors stated “not applicable”. However, in the Results section they are generally describing associations or differences observed in individual studies. Even though many times an effect size may not be estimated, the type of association, the direction of the association and/or the frequency of X outcome between groups is needed. This is noted mainly in the pages and lines described below: Page 17, line 21-24	Thank you, the full details of the quantitative results are reported in Appendix 6. The section on “results of individual studies” has been updated to Appendix 6.
May be more clear to include the number of studies to estimate the OR (or the number of reports from X studies) Page 18	Thank you for your comment. As mentioned earlier, we mapped the quantitative results to qualitative themes (thematically). We used this approach because the outcomes reported in the quantitative studies were heterogenous and no quantitative meta-analysis was possible – this is a similar approach to a similar review (https://journals.plos.org/plosmedicine/article?id=10.1371/journal.pmed.1004074). When we report “quantitative evidence supported qualitative findings”, it means that the results of the quantitative findings are the same, or in the same direction, as qualitative findings. On another hand, “quantitative evidence extended qualitative findings” means that the quantitative findings added a new finding that’s not captured by qualitative findings, but it is still under the same theme. As suggested we have made some changes.
Line 3-5 Important to show the effect of this quantitative analysis. Line 7 “increase the probability of reporting...?” “they will report more frequently.... (X% vs X%)?” Page 18, line 24-25; line 36-38 Numbers are necessary;	Thank you for your comment. As mentioned earlier, we mapped the quantitative results to qualitative themes (thematically). We used this approach because the outcomes reported in the quantitative studies were heterogenous and no quantitative meta-analysis was possible – this is a similar approach to a similar review (https://journals.plos.org/plosmedicine/article?id=10.1371/journal.pmed.1004074).

“higher symptoms in those taking supplements vs those that were not taking..?” Compliers vs non-compliers Page 19 Line 43-44 Any proportion to report? Proportion of women who want fewer tablets, for example? Line 57-60 It is a description. May include a proportion or effect Page 20 Line 28 Include numbers (OR? %?); “associated with increased adherence..” need frequency, probability. OR? Line 56-57—how much higher? Numbers? Page 21, line 19 How they were associated? Page 22 Line 43-44 Describe the quantitative evidence Line 53-56 Only include results here. This sentence is more an interpretation/discussion than a result.	When we report “quantitative evidence supported qualitative findings”, it means that the results of the quantitative findings are the same, or in the same direction, as qualitative findings. On another hand, “quantitative evidence extended qualitative findings” means that the quantitative findings added a new finding that’s not captured by qualitative findings, but it is still under the same theme. We have not made any changes.
Table 3 Below table 3, almost all paragraph (line 21-25) is repeated in the paragraph above (main text). Just completed all the information in the main text.	Thank you for this suggestion, we have removed it.
Discussion In the Prisma 2020 Checklist, in “Discussion” (23 a: Provide a general interpretation of the results in the context of other evidence) the authors refer “page 18 to interpretation”. However, page 18 is still the results section. Discussion with interpretation of the results starts in page 23. But there is no inclusion of other evidence, only discussion within the evidence selected for the review.	Thank you, we have added this on discussion.
Need evidence to contrast and contextualize your results. What has been done before? Are the findings from this review similar/different to other reviews of studies regarding what barriers and facilitators exist among women in achieving adherence to nutrient supplementation? Difficulties in implementation? Health care provider barriers/facilitators?	Thank you, we have added this on discussion. We also followed the Cochrane EPOC template for reporting discussion (https://journals.sagepub.com/doi/pdf/10.1177/16094069211041959).

Evidence from multiple micronutrient supplementation (MMS) studies has reported women’s skills, knowledge, motivation, attitudes, among other themes regarding supplementation. Iron and folic acid supplementation studies also have reported motivators/barriers. Any similarities or differences in the barriers/facilitators that have been reported in other nutrient supplementation implementation experiences in LMICs? In antenatal care? Some suggestions of reviews (Garcia-Casal M.N, Maternal and Child Nutrition, 2018).	
In the second paragraph: The authors mention the need for experience in screening women for high risk of preeclampsia. May be interesting to add reliable and practical options of this screening for LMICs. Also, add some ideas on what experiences exist about assessing low Ca intake?	Thank you for explaining this more. “There is lack of acceptable biomarkers of calcium intake and calcium status, which complicates screening individuals.(18) WHO recommendations on calcium supplementation were set for populations with low calcium intake, as dietary assessments are more reliable to identify populations with low calcium intake rather than to identify individuals. (18)”
Limitations: Any limitations of the qualitative studies? Considering main results are derived from this study design.	Thank you, we have already mentioned the limitation on transferability to other settings.
Figures I cannot see the title of Figure 2.	We have added the following the title and placeholder for both Figure 1 and Figure 2.
Appendix 2 In “Appraisal Results” the authors left it blank. Please complete.	We have completed this item with “Appendix 4”.

VERSION 2 – REVIEW

REVIEWER	Robalino, Shannon Oregon Health & Science University School of Medicine, Center for Evidence-based Policy
REVIEW RETURNED	28-Sep-2023

GENERAL COMMENTS	Feedback Related to Appendices  Appendix 1 – PRISMA Checklist  • Review the items to make sure you have reported them  ○ Example: Item 13f directs the reader to a section that does not exist (and is not reported elsewhere), but also is not applicable • Review the “Location” column to make sure these reflect the locations in your manuscript  ○ Example: Item 14, the location “Methods – Data management, analysis, and synthesis” does not exist • Check references to appendices  ○ Example: Item 7, says Methods – Appendix 1, but should read Appendix 3 Appendix 2 – ENTREQ Checklist  • Similar comments to Appendix 1 Appendix 3 – Search Strategy  • Add platform to each strategy/database (e.g., Ovid)  ○ Example: Embase via Ovid (inception to 2021 March 22) Appendix 4 – Quantitative Studies  • You will likely need to revise this table as it is unnecessary to assess each publication from the same study. This will lead to amendments in Appendix 5. • Define grey cells. Are they “No” or “not applicable”? Appendix 5 – Table 2. Summary of qualitative findings  • This is nearly identical to Appendix 6. Appendix 6 is clearer. I suggest removing Appendix 5 and provide a very brief summary in the manuscript (see comments there) and refer to Appendix 6 for full details. Appendix 6 – Evidence Profile  • Great table! Easy to read. • Suggest renaming to Summary of qualitative findings Appendix 7 – COM-B Mapping Table  • Suggest changing either red or green to another color for accessibility (red/green colorblindness)
---

VERSION 2 – AUTHOR RESPONSE

Reviewer: 1 [please see **two**** attached files for additional comments from reviewer 1]**

Ms. Shannon Robalino, Oregon Health & Science University School of Medicine

General comments to the Author:

Thank you for revising and resubmitting your manuscript. Upon reading your responses to feedback from me and the other reviewer and re-reading the revised manuscript, I feel there are major issues with the way the publications have been synthesized. Publications have been misappropriated to the MICa trial (and may be the Alive & Thrive trial). Disentangling these publications may not result in major changes to your results, but is necessary and will require revisions throughout the manuscript. There are some minor issues with some of the Appendices which I have noted in a separate document.

Response: Thank you, we have tried addressing the reviewer suggestions below.

Feedback Related to Appendices

Appendix 1 – PRISMA Checklist

Review the items to make sure you have reported them

Example: Item 13f directs the reader to a section that does not exist (and is not reported elsewhere), but also is not applicable

Response: Thank you, we have re-checked throughout and modified 13f it to “not applicable”.

Review the “Location” column to make sure these reflect the locations in your manuscript

Example: Item 14, the location “Methods – Data management, analysis, and synthesis” does not exist

Response: Thank you. We have modified the location name.

Check references to appendices

Response: We have checked references and modified them to include Appendix 5 and reordered Appendices.

Example: Item 7, says Methods – Appendix 1, but should read Appendix 3

Response: We have modified the location to Appendix 3.

Appendix 2 – ENTREQ Checklist
Similar comments to Appendix 1

Response: Thank you. We have modified location names.

Appendix 3 – Search Strategy
Add a platform to each strategy/database (e.g., Ovid)

o Example: Embase via Ovid (inception to 2021 March 22)

Response: Thank you. We have added the platform on search strategy.

Methods: Search methods: “CINAHL and Global Health via EBSCO”

Appendix 4 – Quantitative Studies

You will likely need to revise this table as it is unnecessary to assess each publication from the same study. This will lead to amendments in Appendix 5.

Define grey cells. Are they “No” or “not applicable”?

Response: Thank you, we have re-checked all the studies, and we found that there are four publications coming from two studies: (1) Omatayo 2018b & Martin 2017c (quantitative study), and (2) Martin 2017b & Omatoyo 2018a (mixed methods study). For these two publications, we have merged the critical appraisal into one critical appraisal and merged in Table 1 characteristics of study too. For the other publications, we have decided that if the study comes from the same project, yet the publications reported any difference in sample sizes, participant characteristics, settings, and methods, we will not merge it. This is because we feel it is not appropriate to merge and assess them as one study, despite they are coming from the same project. For example, the two studies from the MICA project listed below use different study designs and involve different participants: one qualitative process evaluation with providers and women, while the other one is a qualitative study with providers, women, and adherence partners, conducted at different times. So, we feel it makes more sense to critically appraise these publications separately instead of one. It will be logical, however, to merge the critical appraisal, if there are publications coming from the same methods, same participants, same time frame, and same settings as Omatoyo 2018b and Martin 2017c. We hope this is fine. Yes, grey cells are “Not applicable”, we have clarified this below the table.

Studies:

Martin SL, Seim GL, Wawire S, Chapeau GM, Young SL, Dickin KL. Translating formative research findings into a behaviour change strategy to promote antenatal calcium and iron and folic acid supplementation in western Kenya. *Matern Child Nutr.* 2017 Jan;13(1):10.1111/mcn.12233. doi: 10.1111/mcn.12233. Epub 2016 Feb 22. PMID: 26898417; PMCID: PMC6866120.

Martin SL, Wawire V, Ombunda H, Li T, Sklar K, Tzehaie H, Wong A, Pelto GH, Omotayo MO, Chapleau GM, Stoltzfus RJ, Dickin KL. Integrating Calcium Supplementation into Facility-Based Antenatal Care Services in Western Kenya: A Qualitative Process Evaluation to Identify Implementation Barriers and Facilitators. *Curr Dev Nutr*. 2018 Aug 23;2(11):nzy068. doi: 10.1093/cdn/nzy068. PMID: 30402593; PMCID: PMC6215767.

Appendix 5 – Table 2. Summary of qualitative findings

This is nearly identical to Appendix 6. Appendix 6 is clearer. I suggest removing Appendix 5 and provide a very brief summary in the manuscript (see comments there) and refer to Appendix 6 for full details.

Response: Thank you, based on [A8] comment below and also discussion within the team (as a summary of qualitative findings is the standard reporting for GRADE-CERQual), we have returned Table 2 to the main manuscript document instead of the appendix, and we have renamed appendix 6 to 5 and so on. We have also modified table 2 to fit 2 pages at maximum.

Appendix 6 – Evidence Profile

Great table! Easy to read.

Suggest renaming to Summary of qualitative findings

Response: Thank you. Unfortunately, per the GRADE-CERQual convention, this table is called the evidence profile, while Table 2 is the summary of the qualitative findings table. We want to be consistent with the naming convention to ensure fidelity to the approach. No change is made on this.

Appendix 7 – COM-B Mapping Table

Suggest changing either red or green to another color for accessibility (red/green colorblindness)

Response: Thank you we have changed the colors and uploaded a new file.

Feedback on the Main Manuscript Document

Commented [A1]: *No other mention of risk of bias in manuscript. In the Methods section you use “quality appraisal”. I’ve suggested some wording in the methods section.*

Response: Thank you for the suggestion.

Commented [A2]: *Were these also in Ovid? Or another platform? For example, Medline and Embase are databases and Ovid is a platform.*

Response: Thank you, we have added this information to the manuscript, CINAHL and Global Health via EBSCO. See previous comment.

Commented [A3]: *Thank you for your response regarding the involvement of an experienced librarian in the development and execution of the search strategy. However, I will reiterate that the use of the Humans Only limit is inappropriate for a systematic review because it will miss studies that were recently published (up to 18 months prior to the search date). I provided a list of 17 citations which you reviewed, but there were likely many more.*

Response: Thank you. We will take into account that the exclusion of humans only limits for the future reviews that we conduct.

Commented [A4]: I do not see Figure 1 in the manuscript or appendices. Is this the PRISMA diagram?

Response: Yes, it is PRISMA diagram, The Figure has been uploaded to the system, and the title and legends can be seen after references.

Commented [A5]: *As previously mentioned, it is unusual and misleading to separate and report publications from a single study this way. However, before revising, please see comments for Table 1.*

Response: Thank you we have revised this throughout based on the merging of four publications into two studies.

Commented [A6]: *Insert references as appropriate.*

This table is not a table of study characteristics; instead this is a table of the characteristics of each paper cited. Study characteristics should be reported by study, not each individual paper as this makes it appear as though there were more participants than there were. You state you included 9 studies in 18 papers so there should be 9 rows maximum though you can provide notes of other publications from the same study. Given the majority of studies are independent, it may be prudent to rearrange this table by Author and Year or Study Name in the first column. This is a good example of study characteristics table <https://journals.plos.org/plosone/article?id=10.1371/journa>.

I.pone.0227765#pone-0227765-t001 .

Additionally, I believe many of these papers are misattributed to the MICa trial (NCT02238704) because of overlapping authors. The MICa trial (as indicated by the trial registration) is an RCT that enrolled 1,032 pregnant women. I strongly suggest you closely re-examine each paper in this table/review and group them together as appropriate.

For example:

- Birhanu 2018 (19; mislabelled as 2016 in the table) not part of the MICa trial*
- Martin 2017 (38; mislabelled as 2016 in the table) not part of the MICa trial*
- Martin 2018 (39) is part of the MICa trial as stated in the introduction “The Micronutrient Initiative-Cornell University Calcium (MICa) Trial (NCT02238704) was a cluster- randomized noninferiority trial that examined the effect of supplementation regimen on the amount of calcium supplement ingested (5, 11), and included a nested qualitative process evaluation.” See also details in the Process Evaluation section.*

Response: Thank you we have tried addressing this, please see Table 1 revision. Regarding the MICA trial, we have emailed the confirmed with the authors that the publications we listed as MICA trial are indeed not coming from the MICA trial, but they are part of MICA project. So, we have mentioned this as a MICA project throughout the manuscript instead of a trial. Please note that we have removed the project names on Table 1 as suggested based on the example.

Commented [A7]: You will need to amend the results section throughout after you disentangle the issues noted in Table 1.

I maintain the use of the parentheticals at the end of the sentence is confusing. If you feel it is important to keep that information, I suggest moving it to earlier in the sentence. For example:

Three studies (in 5 papers) aimed to evaluate the implementation of calcium supplements in pregnancy.

Response: Thank you we have revised this throughout based on the merging of four publications into two studies.

Commented [A8]: *It is worthwhile keeping a brief summary table in the main body of the manuscript. You can state the full details are found in Appendix 5. See example*

Explanation of overall assessment *not in CERQual in Table 2 of this 2018 document*
<https://drive.google.com/file/d/1tF0rhDmihpzd3B6BDFHcD>

[8msDu1rqYjR/view](https://drive.google.com/file/d/1tF0rhDmihpzd3B6BDFHcD)

Response: Thank you, we have returned Table 2 to the main manuscript.

VERSION 3 – REVIEW

REVIEWER	Robalino, Shannon Oregon Health & Science University School of Medicine, Center for Evidence-based Policy
REVIEW RETURNED	27-Nov-2023
GENERAL COMMENTS	Thank you for the revisions. Your paper is in great shape. Only 1 minor comment that relates to Table 1. Make sure you are presenting the studies in the table consistently. For example, in the Qualitative grouping you have ordered the papers alphabetically by author, but in the Quantitative grouping they aren't in any sort of order.

VERSION 3 – AUTHOR RESPONSE

Reviewer: 1

Ms. Shannon Robalino, Oregon Health & Science University School of Medicine

Comments to the Author:

Thank you for the revisions. Your paper is in great shape. Only 1 minor comment that relates to Table 1. Make sure you are presenting the studies in the table consistently. For example, in the Qualitative grouping you have ordered the papers alphabetically by author, but in the Quantitative grouping they aren't in any sort of order.

Response: Thank you for this, we have updated the order of studies.

Reviewer: 1

Competing interests of Reviewer: None.